

# Ornamentation of dermal bones of *Metoposaurus krasiejowensis* and its ecological implications

Mateusz Antczak[1] and Adam Bodzioch[2]

[1] Institute of Geology, Adam Mickiewicz University of Poznan, Poznan, Poland
[2] Department of Biosystematics, University of Opole, Opole, Poland

## ABSTRACT

**Background**. Amphibians are animals strongly dependent on environmental conditions, like temperature, water accessibility, and the trophic state of the reservoirs. Thus, they can be used in modern palaeoenvironmental analysis, reflecting ecological condition of the biotope.

**Methods**. To analyse the observed diversity in the temnospondyl *Metoposaurus krasiejowensis* from Late Triassic deposits in Krasiejów (Opole Voivodeship, Poland), the characteristics of the ornamentation (such as grooves, ridges, tubercules) of 25 clavicles and 13 skulls were observed on macro- and microscales, including the use of a scanning electron microscope for high magnification. The different ornamentation patterns found in these bones have been used for taxonomical and ecological studies of inter- vs. intraspecific variation.

**Results**. Two distinct types of ornamentation (fine, regular and sparse, or coarse, irregular and dense) were found, indicating either taxonomical, ecological, individual, or ontogenetic variation, or sexual dimorphism in *M. krasiejowensis*.

**Discussion**. Analogies with modern Anura and Urodela, along to previous studies on temnospondyls amphibians and the geology of the Krasiejów site suggest that the differences found are rather intraspecific and may suggest ecological adaptations. Sexual dimorphism and ontogeny cannot be undoubtedly excluded, but ecological variation between populations of different environments or facultative neoteny (paedomorphism) in part of the population (with types of ornamentations being adaptations to a more aquatic or a more terrestrial lifestyle) are the most plausible explanations.

## INTRODUCTION

The fossil assemblage from the Late Triassic deposits in Krasiejów (SW Poland, near the city of Opole) is a unique discovery. Excavations carried out since 2000 have revealed new data concerning the evolution of terrestrial Triassic faunas. In Krasiejów, although the remains of several groups of fish and archosaurs were also found (e.g., *Dzik & Sulej, 2007*; *Dzik & Sulej, 2016*; *Brusatte et al., 2009*; *Piechowski & Dzik, 2010*; *Sulej, 2010*; *Skrzycki, 2015*; *Antczak, 2016*; *Dzik & Sulej, 2016*; *Antczak & Bodzioch, 2018*), fossils of large temnospondyl

Corresponding author
Mateusz Antczak,
mateusz.antczak@amu.edu.pl

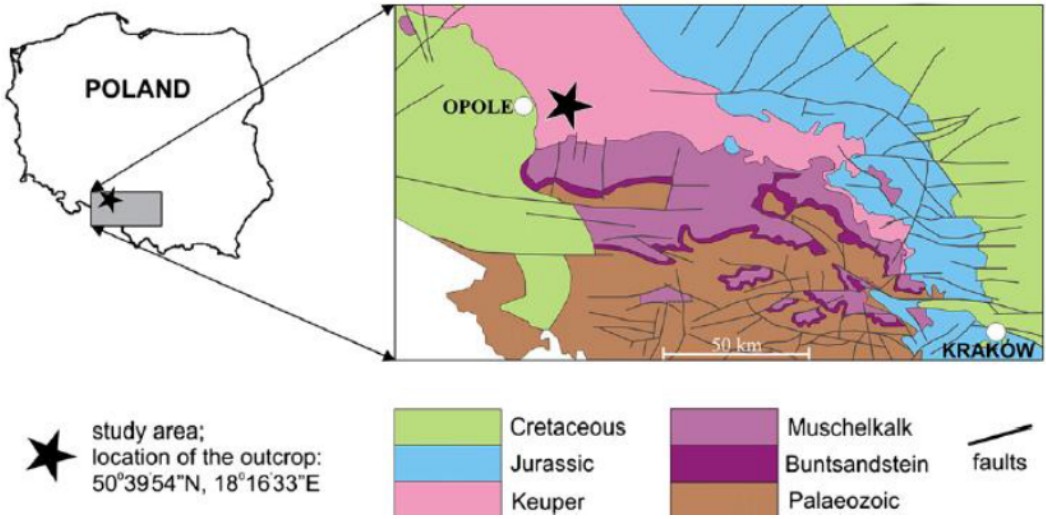

**Figure 1** Localization and geological map of Krasiejów (simplified fragment after *Dadlez, Marek & Pokorski, 2000*).

amphibians described as *Metoposaurus krasiejowensis* (*Sulej, 2002*; species name revised by *Brusatte et al., 2015*) were the most abundant.

Despite many years of study, new data are still being collected and some aspects of the anatomy and ecology of extinct animals are being reinterpreted (e.g., *Konietzko-Meier & Klein, 2013*; *Konietzko-Meier & Sander, 2013*; *Gruntmejer, Konietzko-Meier & Bodzioch, 2016*; *Konietzko-Meier et al., 2018*), along with the age of bone accumulations in Krasiejów (*Lucas, 2015*; *Szulc, Racki & Jewuła, 2015*) and their origin (*Bodzioch & Kowal-Linka, 2012*). One aspect not described in detail is the morphology of metoposaurid dermal bone ornamentation, which was assumed to be randomly variable (*Sulej, 2007*) or similar in all representatives of the species, as suggested by *Witzmann et al. (2010)*. The aim of this paper is to describe in detail, on macro- and microscales, the ornamentation of metoposaurid clavicles and skull bones, in order to examine its variation and to test whether or not it is the same in all specimens. A thorough probe of skeletal elements from one site shows that differences between specimens are not random.

## MATERIAL AND METHODS

The size, number, shape, placement, and other features of the ornamentation elements of metoposaurid clavicles (and as a remark: skull bones) were analysed. The material derived from the 'Trias' site at Krasiejów (SW Poland; Fig. 1). The fine-grained (mudstones and claystones) Late Triassic (Carnian, according to *Dzik & Sulej, 2007*; *Lucas, 2015*; Norian, according to *Szulc, 2005*; *Szulc, 2007*; *Szulc, Racki & Jewuła, 2015*) deposits can be divided into three units (e.g., *Gruszka & Zieliński, 2008*), in which two bone-bearing horizons occur. The lower horizon, the product of a mudflow deposition that probably occurred during a heavy rainy season, is especially abundant in fossils, including *M. krasiejowensis*. The upper horizon was described within massive claystones covering palaeochannels of low-energy

meandering river. Within the upper horizon remains of *Silesaurus* and *Polonosuchus* were found (*Dzik & Sulej, 2007*).

To test the diversity of dermal bone ornamentation in metoposaurids from Krasiejów, 25 clavicles (UOPB1152–1176) and 13 skulls (working numbers counting from the excavation site: UO/PP01–20) were analysed in detail (Tables 1–3). Morphometric measurements for 21 skulls were also taken (Table 4). The clavicles were removed during the excavation and are held in the Opole University collection, while the skulls were present *in situ* in the palaeontological pavilion (also part of Opole University) at the digging site in Krasiejów; one of them is housed in the Faculty of Geographical and Geological Sciences Museum of Earth at the Adam Mickiewicz University in Poznań (uam/mz/586). As an outgroup skull and clavicle of *Cyclotosaurus* (ZPAL/AbIII/397) from Museum of evolution in Warsaw were examined.

All described specimens were found in the lower bone-bearing horizon.

The characteristics of the polygonal and radial structure of clavicles were described, using over 20 features, including some of the 12 used by *Witzmann et al. (2010)*. Observations are shown in Table 1, which groups similar features and assigns them numerical values.

Observations were made macroscopically and microscopically using an Olympus SZ61 binocular microscope, a Zeiss SteREO microscope, and a DIGEYE digital microscope.

Fragments of 10 clavicles were analysed using a Hitachi S-3000N Scanning Electron Microscope. Samples were taken from the same parts of the clavicles: radial ornamentation in the posterior region of the bone, several centimetres behind the ossification centre. Samples were sprayed with gold and palladium and observed under a high vacuum at the Institute of Plant Protection − National Research Institute in Poznań. One sample was observed using a Hitachi S-3700N at the SEM-EDS Laboratory of Faculty of Geographical and Geological Science of Adam Mickiewicz University in Poznań.

Selected macroscopic features of skull bones were described only as a result of the fact that the presentation of bones *in situ* makes it impossible to describe micro- or sub-microscopic features. Not all such features were described. Dermal bone ornamentation can be divided into radial ornamentation, composed of parallel or radial ridges without transverse ridges, and polygonal ornamentation, composed of short ridges connected to form polygons. The vertices of the polygons are called nodal points. The polygonal sculpture area is the ossification centre, the part of the bone that ossifies first. Near the ossification centre is an anterior appendix. Polygons may be hexagonal, pentagonal, rectangular, or irregular in shape. Polygons joined by means of a missing ridge are called multipolygons (Fig. 2). All measured features are listed in Table 1. SEM observations included features of the surface of the ridges, such as the number of foramina and degree of ridge roughness (Fig. 3). The possible relative individual ages of the clavicle specimens were determined using the method based on ornament development, presented by *Witzmann et al. (2010)* and improved by Zalecka (K Zalecka, 2012, unpublished data).

For testing the significance of described variation statistical test were used. At first Shapiro–Wilk test for testing normality of the data, then respectively test F, test T and test U. Test F was used if both compared samples had normal distribution. Test F was used for testing the variance. If the difference between variances were not significant, test T was

Antczak and Bodzioch (2018), *PeerJ*, DOI 10.7717/peerj.5267

**Table 1 Clavicles ornamentation.**

| | Zpal AbIII 397 | UOPB 1152 | UOPB 1153 | UOPB 1154 | UOPB 1155 | UOPB 1156 | UOPB 1157 | UOPB 1158 | UOPB 1159 | UOPB 1160 | UOPB 1161 | UOPB 1162 | UOPB 1163 | UOPB 1164 | UOPB 1165 | UOPB 1166 | UOPB 1167 | UOPB 1168 | UOPB 1169 | UOPB 1170 | UOPB 1171 | UOPB 1172 | UOPB 1173 | UOPB 1174 | UOPB 1175 | UOPB 1176 |
|---|---|---|---|---|---|---|---|---|---|---|---|---|---|---|---|---|---|---|---|---|---|---|---|---|---|---|
| | *Cycl* | T1 | T2 | T2 | T2 | T2 | T2 | T1 | T1 | T1 | T1 | T1 | T2 | T2 | T2 | T1 | T1 | T2 | T2 | T1 | T1 | T2 | T1 | T1 | T2 | T1 |
| Age (K Zalecka, 2012, unpublished data): juvenile (J), intermediate (I), adult (A) | | A | I | J | J | | | | J | J/I | | | | A | J/I | I | J | I | J | J | **I** | I | I | I | A | J |
| Regular (1), irregular (2) | 1 | 1 | 2 | 2 | 2 | 2 | 2 | 1 | | 1 | 1 | 1 | | 2 | 2 | 1 | 1 | 2 | 2 | 1 | 1 | 2 | 1 | 1 | 2 | 1 |
| Very fine (0), fine (1), coarse (2), very coarse (3) | 3 | 1 | 1 | 2 | 2 | 2 | 2 | 1 | 1 | 1 | 2 | 2 | | 1 | 2 | 1 | 0 | 2 | 2 | 0 | 1 | 2 | 1 | 1 | 2 | 1 |
| Very sparse (0), sparse (1), dense (2) | 0 | 1 | 2 | 2 | 2 | 1 | 2 | 1 | 1 | 1 | 1 | 1 | | 2 | 2 | 1 | 2 | 2 | 2 | 2 | 1 | 1 | 2 | 1 | 2 | 1 |
| Av. polygon diameter/av. ridge width [<4 (1), >4 (2), >6 (3)] | 3 | 2 | 1 | | 1 | 1 | 1 | 2 | | 2 | 2 | 2 | | 1 | 1 | 2 | 1 | | 1 | 1 | 1 | 1 | | | | |
| Distinct borders of polygonal field (1), borders partially hard to recognize (2), hard to recognize (3) | 3 | 1 | 2 | | 3 | | | | | 1 | 1 | 1 | | 3 | | 1 | 1 | | 2 | 1 | 1 | 3 | 1 | | 3 | 1 |
| Ridge quantity/bone width [measurement 2,5 cm from polygon border]: >2,3 (2), <2,3 (1), <2 (0) | 0 | 1 | 2 | 2 | | | | | 1 | | 1 | 1 | | 2 | 2 | 1 | | 2 | 2 | 1 | 1 | | | | | |
| Nodal points slightly wider than ridges (1); some nodal points distinctly wider than ridges (2); nodal points distinctly wider than ridges (3) [*Witzmann et al., 2010*] | 3 | 3 | 2 | 1 | 1 | 1 | 1 | 3 | | 3 | 3 | 2 | | 2 | 2 | 3 | 3 | 1 | 1 | 3 | 3 | 1 | 1 | 3 | 1 | 3 |
| Ridges edged (1); round or edged (2); round (3) [*Witzmann et al., 2010*] | 3 | 1 | 2 | 3 | 3 | 3 | 2 | 1 | 2 | 2 | 2 | 2 | 3 | 2 | | 1 | 1 | 3 | 3 | 2 | 2 | 2 | 2 | | | |
| Deep polygons (1),deep or shallow polygons (2); shallow polygons (3) | 1 | 1 | 1 | 3 | 3 | 3 | 3 | 1 | | 1 | 1 | 2 | | 2 | 1 | 1 | 1 | 2 | | | 1 | 2 | | | | |
| Polygon shape:>50% hexagons (1), <50% hexagons (2), >50% quadrangle (3) | 3 | 1 | 2 | 2 | 2 | 2 | 2 | 1 | | 1 | 1 | 1 | | 2 | 2 | 1 | 1 | 2 | 2 | 1 | 2 | 2 | 1 | 1 | 2 | 1 |
| Polygon size: usually small (1), usually large (2), very large (3) [large: >0,4 mm diameter]. | 3 | 1 | 2 | 2 | 2 | 2 | 2 | 1 | | 1 | 1 | 1 | | 2 | 1 | 1 | 1 | 2 | 2 | 2 | 1 | 2 | 1 | 1 | 2 | 1 |
| Multipolygons: several or none (1), numerous(2) [more than 11] | 1 | 1 | 2 | 2 | 2 | 1 | 2 | 1 | | 1 | 1 | 1 | | 2 | 2 | 1 | 1 | 2 | 2 | 1 | 1 | 2 | 1 | 1 | 2 | 1 |
| Polygon field shape: square (1), rectangular (2), elongated (3) | 3 | 1 | 2 | | 2 | | | 1 | | 1 | | | | 2 | | 1 | 1 | | 2 | | 1 | 2 | 1 | | | 1 |

*(continued on next page)*
## Table 1 (*continued*)

| | Zpal AbIII 397 | UOPB 1152 | UOPB 1153 | UOPB 1154 | UOPB 1155 | UOPB 1156 | UOPB 1157 | UOPB 1158 | UOPB 1159 | UOPB 1160 | UOPB 1161 | UOPB 1162 | UOPB 1163 | UOPB 1164 | UOPB 1165 | UOPB 1166 | UOPB 1167 | UOPB 1168 | UOPB 1169 | UOPB 1170 | UOPB 1171 | UOPB 1172 | UOPB 1173 | UOPB 1174 | UOPB 1175 | UOPB 1176 |
|---|---|---|---|---|---|---|---|---|---|---|---|---|---|---|---|---|---|---|---|---|---|---|---|---|---|---|
| | *Cycl* | T1 | T2 | T2 | T2 | T2 | T2 | T1 | T1 | T1 | T1 | T1 | T2 | T2 | T2 | T1 | T1 | T2 | T2 | T1 | T1 | T2 | T1 | T1 | T2 | T1 |
| Ridge height: lower than nodal points (1), almost equal to nodal points (2) | | 1 | 2 | 2 | 2 | 2 | 2 | 1 | | 1 | 1 | 2 | 2 | 2 | 1 | 1 | 2 | 1 | 2 | 1 | 1 | 2 | 1 | | 2 | |
| Ossification degree: low (1), high (thick bones) (2) | 2 | 1 | 2 | 2 | 2 | 2 | | | 1 | 1 | 2 | 1 | | 2 | 2 | 1 | 2 | 2 | 2 | 2 | 1 | 2 | | 2 | 2 | 1 |
| Anterior clavicle projection: small and flat (1), round and expanded (2), more than 45 deg. (3) | 3 | 1 | 2 | 2 | 2 | 2 | | | 1 | 1 | | | 2 | 2 | | 1 | 2 | 2 | 2 | 1 | 2 | 2 | | | | |
| More ramifications: opening (1), closing (2) | | 1 | 1 | 1 | 1 | | | 2 | | | | | | 1 | | | | | | | | | | | | |
| Shape of the radial ridges: undulated (1), straight (2) | 2 | 2 | 2 | 2 | 2 | | | | 2 | | | 2 | 2 | 2 | 2 | | | | | | | | | | | |
| Ridge surface (macroscale): bumps (1), large cuts (2), small cuts (3) | | 1 | 2 | 2 | 3 | | 2 | 3 | 3 | 3 | 3 | | | 1 | | | | | | | | | | | | |
| Ridge width<half of the polygon diameter: yes (1), no (2) (*Witzmann et al., 2010*) | | 1 | 1 | | 2 | 1 | 1 | 1 | | | 1 | 1 | | 1 | 1 | | | | | | | | | | | |
| Radial ridges constrictions and height differences: distinct (1), not distinct (2) | | 1 | 1 | 2 | 2 | 2 | | | 1 | 1 | 1 | 2 | 1 | 1 | 1 | 1 | 2 | 2 | 2 | | | 2 | 1 | | | |
| Shape of the clavicle (angle) >**100°** (1), <**100°** (2) | 2 | 1 | | 2 | 2 | | | | | | 1 | | | 2 | | | 1 | | | 1 | | 2 | | | | |

Antczak and Bodzioch (2018), *PeerJ*, DOI 10.7717/peerj.5267

**Table 2  SEM observations of the clavicles.**

| | UOPB 1152 | UOPB 1153 | UOPB 1155 | UOPB 1157 | UOPB 1160 | UOPB 1161 | UOPB 1163 | UOPB 1164 | UOPB 1167 | UOPB 1168 | UOPB 1169 |
|---|---|---|---|---|---|---|---|---|---|---|---|
| **Roughness** [v – distinct x – not distinct] | x | v | v | v | v | v | v | v | x | v | v |
| **Striations** [v – distinct, numerous x – few] | x | v | v | v | x | x | v | | x | v | v |
| **Small foramina** [v – more than 7/100 $um^2$ x – less than 7/100 $um^2$] | x | v | v | v | x | x | v | v | x | v | v |
| **Large foramina** [v – more than1/1 mm of length x – less than 1/1 mm of length] | x | x | v | v | x | v | v | v | x | v | x |

Antczak and Bodzioch (2018), *PeerJ*, DOI 10.7717/peerj.5267

| Table 3 | Skulls ornamentation. | | | | | | | | | | | | | | |
|---|---|---|---|---|---|---|---|---|---|---|---|---|---|---|---|
| | | UO/PP01 | UO/PP02 | UO/PP04 | UO/PP06 | UO/PP08 | UO/PP09 | UO/PP12 | UO/PP13 | UO/PP14 | UO/PP16 | UO/PP17 | UO/PP18 | UO/PP20 | uam/mz/586 |
| Parietal-supratemporal ornament | Mostly: polygons (2), radial ridges (1) | | 2 | 1 | 1 | 2 | 1 | 2 | 1 | 1 | 2 | 1 | 2 | 2 | 1 |
| Postfrontal-postorbital ornament | Mostly: polygons (3), Polygons and radial ridges (2), radial ridges (1) | | 3 | 2 | 1 | 3 | 2 | 3 | 2 | | 2 | 2 | 3 | 3 | 2 |
| Squamosal ornament | Mostly: polygons (2), radial ridges (1) | 1 | 1 | 1 | | | 1 | 1 | 1 | | 1 | 1 | | | 1 |
| Multipolygons | Occurs (2), not occur (1) | | 2 | 1 | | | | 2 | 1 | | 2 | 2 | | | 1 |
| Polygon shape | Irregular (2), mostly hexagonal (1) | 2 | 2 | 1 | | 2 | 1 | 2 | 1 | 1 | 2 | 1 | 2 | 2 | 1 |
| Polygon size | Small (2), large (1) | | 2 | 2 | | 2 | 1 | 2 | 1 | 2 | 2 | 1 | 2 | 2 | 2 |
| Polygon density | Sparse (1), dense (2) | | 2 | 1 | | 2 | 1 | 2 | 1 | | 2 | | 2 | 2 | 1 |

Antczak and Bodzioch (2018), *PeerJ*, DOI 10.7717/peerj.5267

**Table 4  Skull measurements (in cm).**

| | UO/PP01 | UO/PP02 | UO/PP03 | UO/PP04 | UO/PP05 | UO/PP06 | UO/PP07 | UO/PP08 | UO/PP09 | UO/PP10 | UO/PP11 | UO/PP12 | UO/PP13 | UO/PP14 | UO/PP15 | UO/PP16 | UO/PP17 | UO/PP18 | UO/PP19 | UO/PP20 |
|---|---|---|---|---|---|---|---|---|---|---|---|---|---|---|---|---|---|---|---|---|
| *Skull roof* | | | | | | | | | | | | | | | | | | | | |
| SL | 25.2 | 28.4 | | 35 | | 34 | | 28.8 | 43.1 | | | 28 | 42.7 | 28.5 | | 30.3 | 34.1 | 35.4 | 32.7 | 33 |
| SW | 21 | 24.5 | | 28.6 | | 31 | | ~23 | 36.8 | | | ~25.8 | 37 | 26.7 | | 26.5 | ~26 | 27.2 | 28.6 | ~29 |
| IN | | | | | | 5 | | 4.7 | 6 | | | ~4.5 | 6 | ~4 | | | | 4.4 | 4.1 | 5 |
| IOL | 7.5 | 7.6 | | 9.1 | | 8.4 | | 8.5 | 12 | | | 7.8 | 12.1 | 8 | | 9 | 9 | 8.8 | 9 | 9 |
| AOL | 10 | | | | | 14.2 | | 13 | 13.5 | | | 9.4 | 13.7 | 8.9 | | 10 | 10.3 | 11.1 | 10.1 | 10.5 |
| POL | 16.8 | 16 | | 18.3 | | 14.9 | | 15.9 | 25.8 | | | 16.4 | 21.4 | 13.8 | | 18.4 | 19.2 | 17.9 | 17.4 | 18 |
| SE | 5 | 7.2 | | | | 6.8 | | 6.2 | 10 | | | 6.4 | 10.7 | 8.1 | | 8.2 | 8.5 | 8.8 | 7.5 | 8.3 |
| ME | 7 | 8.5 | | 9.6 | | 8.8 | | 9 | 11.5 | | | | 13.8 | 9.8 | | 11 | 11.6 | 11 | 9.7 | 11.5 |
| NL | | | | | | 2.6 | | 2.1 | 3.6 | | | | 3.9 | | | 3.1 | 3.2 | 3.4 | ~2.8 | 2.2 |
| I (L) | 2 | 2 | | 1.7 | | 1.6 | | 1.8 | 2.5 | | | 1.8 | 2.3 | 1.9 | | 2 | 1.9 | 2.3 | 2.3 | 2 |
| I (P) | 2 | 2 | | 2.2 | | 2.2 | | 2.2 | 3 | | | 2 | 3 | 1.9 | | 1.9 | 2.3 | 2 | 2 | 2.7 |
| M | | 4.3 | | | | 3.9 | | 3 | 4 | | | | 3 | 3.1 | | | 4 | 3.8 | 3.4 | 4.2 |
| NO | 6.1 | 6.3 | | 7.9 | | 7.4 | | | 9.8 | | | 6.5 | 9 | 6.7 | | 6.4 | 7.2 | 7.5 | 7.4 | 7.5 |
| LO | 2.8 | 4 | | 4.6 | | 4 | | 3.7 | 5.6 | | | 3.9 | 5.6 | 36 | | 4 | 4 | 4.6 | 4.3 | 3.4 |
| MW | | 16 | | 19.4 | | 18.6 | | 18 | 24 | | | 15.7 | 23.5 | 17.2 | | 19 | 19.1 | 19 | 18.5 | 19 |
| *Palate* | | | | | | | | | | | | | | | | | | | | |
| LP | | | 30 | | 30.3 | | | | | 32.1 | 33.4 | | | | 30 | | | | | |
| NP. | | | 9.9 | | 10 | | | | | | | | | | 7.9 | | | | | |
| Y | | | 15.4 | | 15.1 | | 14 | | | 14 | 16 | | | | 14.7 | | | | | |
| R | | | 10.6 | | 11.2 | | 11.4 | | | 14.1 | 13.2 | | | | 11.6 | | | | | |
| B | | | 23.5 | | ~27 | | 20.4 | | | 26.7 | 29 | | | | ~24 | | | | | |
| O | | | | | | | | | | | | | | | 4.4 | | | | | |
| E | | | 6 | | | | | | | | | | | | 4.4 | | | | | |
| G | | | 4.6 | | | | | | | | | | | | | | | | | |

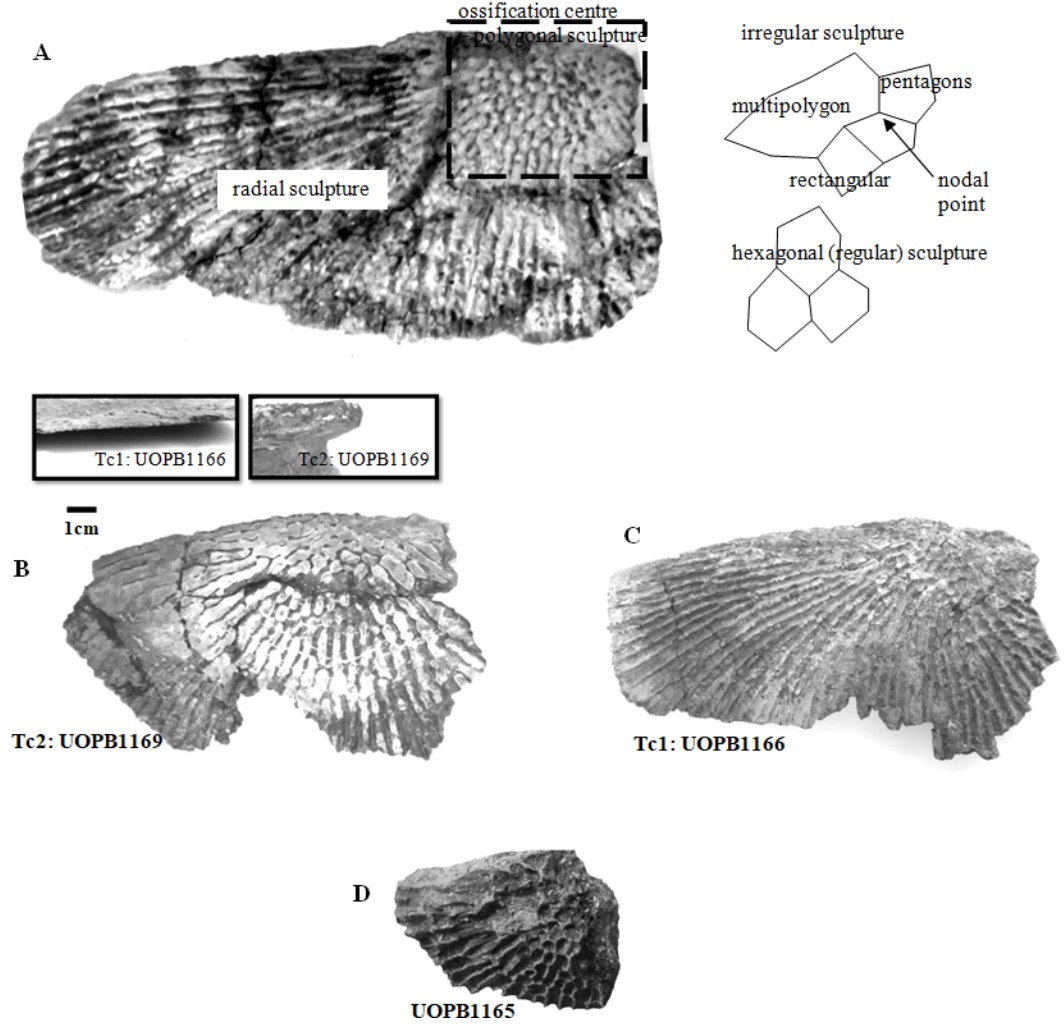

**Figure 2 Clavicles ornamentation character.** (A) Basic ornamentation features explanation. (B) Clavicle assigned to ornamentation type 2 (Tc2). (C) Clavicle assigned to ornamentation type 1 (Tc1). (D) UOBP1165, partially incomplete specimen, not fitting to the described types (C, Figs. 4–6).

used. If the variances were significantly different, or samples not had normal distribution, test U was used. If the final test gave the *p*-value (probability value) less than 0.05 it means that samples are significantly different.

## Observations
### Diagnosis: clavicles

Clavicles of metoposaurids from Krasiejów showed diversity in ornamentation, having fine, regular and sparse, or coarse, irregular and dense sculpture. After this observation, the clavicles ornamentation was examined in greater detail.

Some of the analysed features show random variation or none; however, most are distributed bimodally. Therefore, in every specimen one or the other set of characteristics occur, and two types of ornamentation can be distinguished (Tc1 and Tc2).

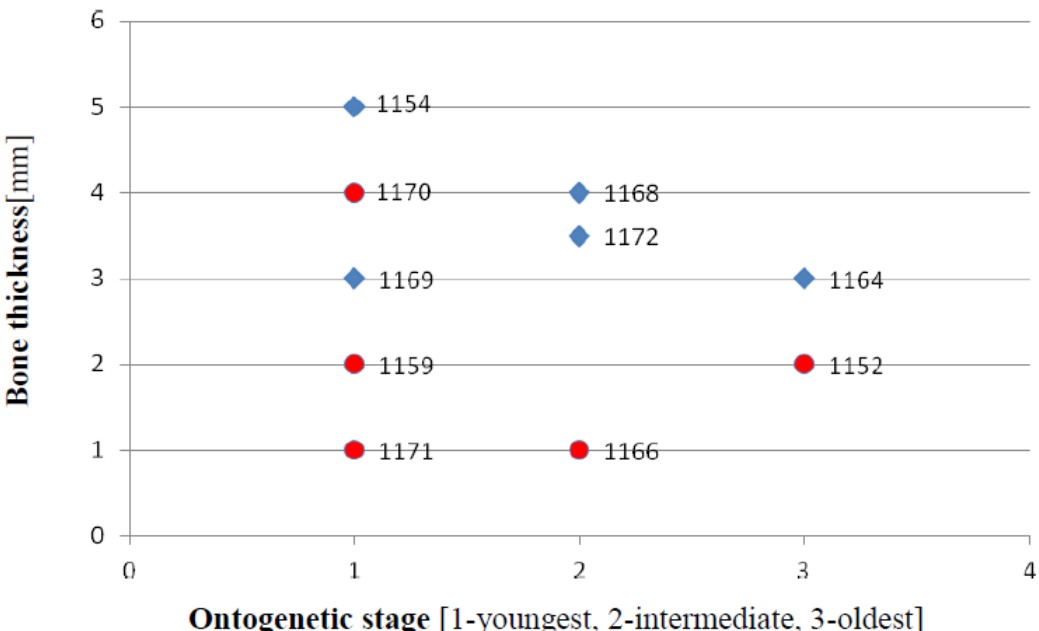

**Figure 3 Thickness of the bone in particular types and ontogenetic stages.** Measurements made at the border of polygonal and radial ornamentation areas.

Specimens classified as type 1 (Tc1) are characterised by more regular ornamentation of the clavicles: the borders of the ossification centre (polygonal sculpture) are easily recognised, the polygonal sculpture field has a square shape, and the ornamentation is fine and sparse, moreover, nodal points are more pronounced, being broader and higher than the ridges that connect them, ridges are usually narrow, hexagons with a low level of size diversity dominate, multipolygons are rare, clavicles, even when large, are relatively thin; the anterior process of the clavicle is usually flat and small (Fig. 2); while specimens classified as type 2 (Tc2) posses less regular ornamentation: the borders of the ossification centre (polygonal sculpture) are difficult to recognise, the polygonal sculpture field is characterised by a rectangular shape (elongated posteriorly), and the ornamentation is thicker and denser, moreover, nodal points are only slightly broader and higher than the ridges that connect them, ridges are wide or narrow, often rounded, polygons are more often pentagonal or irregular, multipolygons are frequent, clavicles are relatively thick,

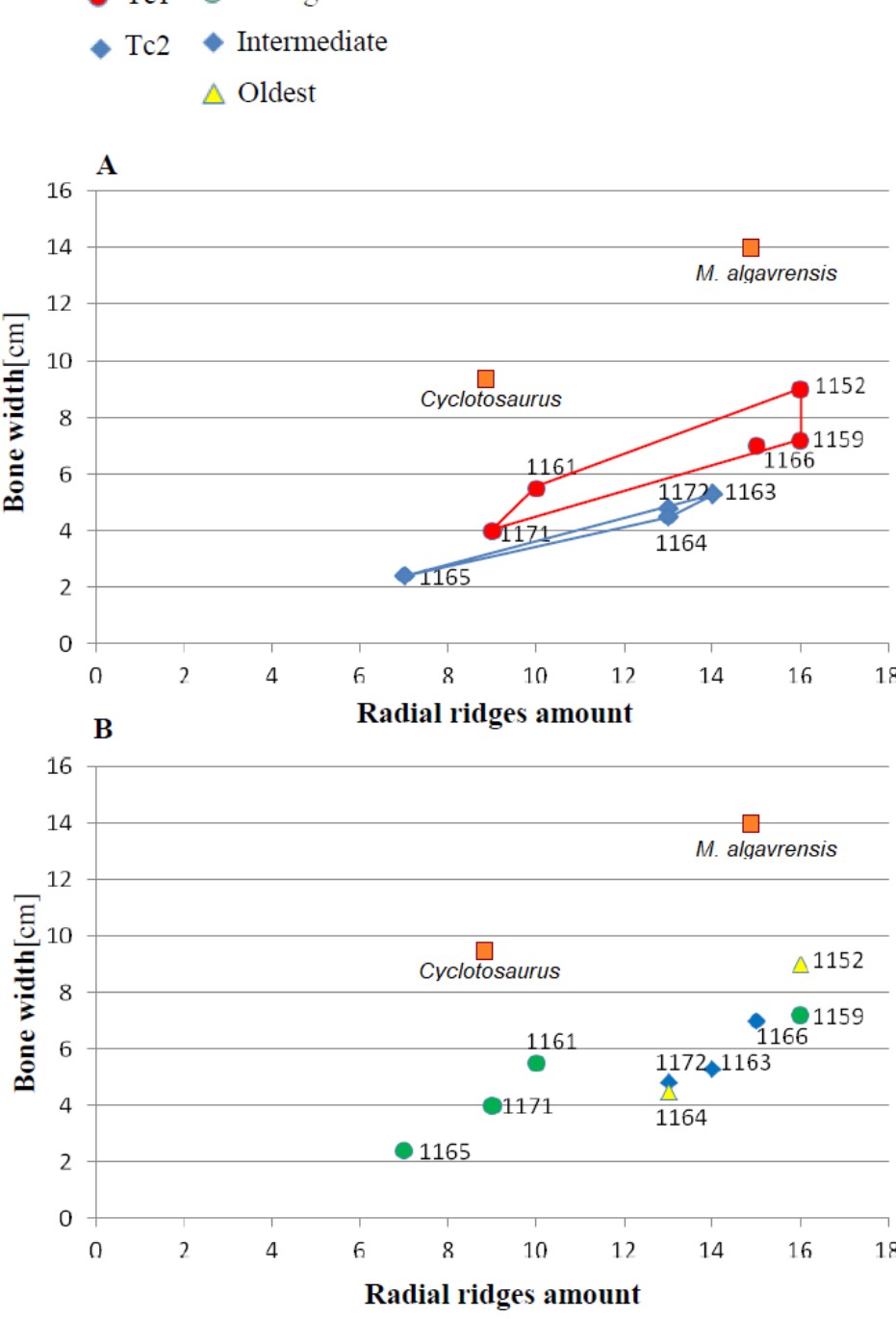

**Figure 4** **Ratio of the bone width and amount of radial ridges.** Measurement taken 2.5 cm, from ossification centre. (A) Considering appointed types, showing two subsets within metoposaurid data. (B) Considering individual age, showing no subsets within metoposaurid data.

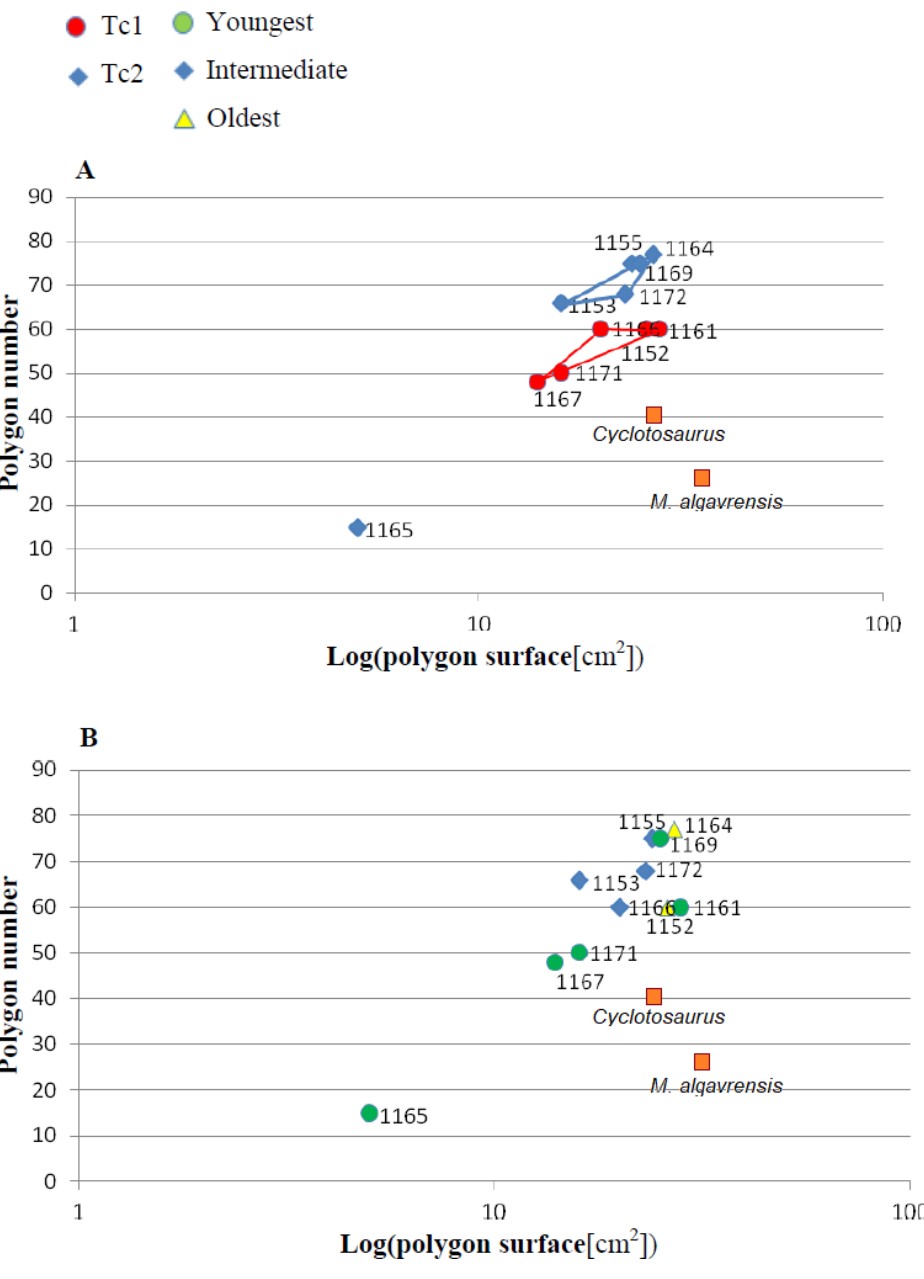

**Figure 5** **Ratio of polygon number and surface.** (A) Considering appointed types, showing two subsets within metoposaurid data. (B) Considering individual age, showing no subsets within metoposaurid data.

independently of their size or age, and the anterior process is usually round in cross section and expanded (Fig. 2).

Both types of *Metoposaurus* dermal bones ornamentation are however distinct from *Cyclotosaurus* sculpture (*Sulej & Majer, 2005*). *Cyclotosaurus* can be characterised by relatively large and rhomboidal polygons (sometimes elongated pentagons). Radial ornament is very sparse (spaces between ridges are wide). Ossification centre is large

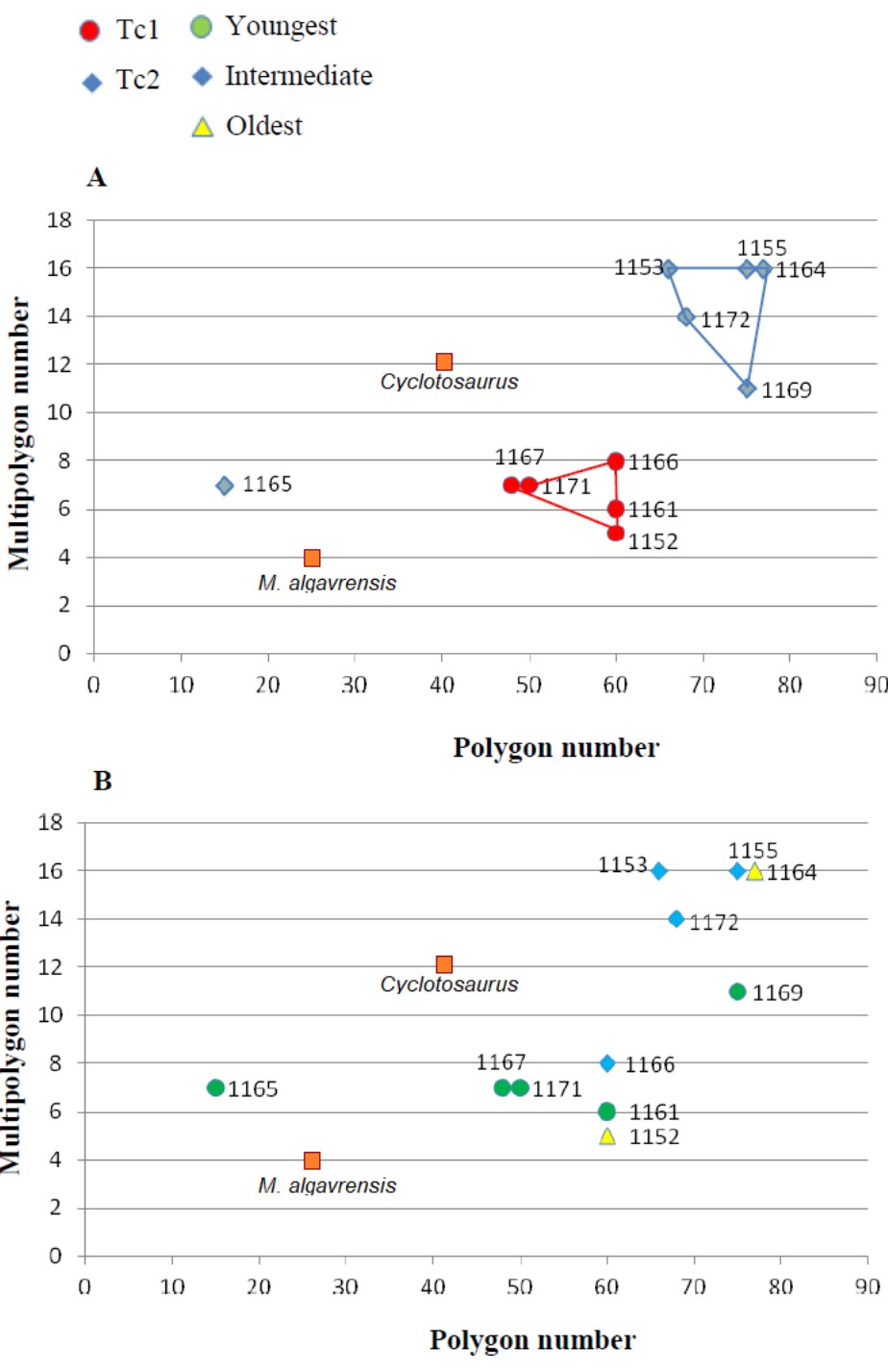

**Figure 6  Ratio of multipolygon and all polygon numbers.** (A) Considering appointed types, showing two subsets within metoposaurid data. (B) Considering individual age, showing no subsets within metoposaurid data.

and posses distinct borders, but the polygon number is low (25). Clavicle is thick. Ridges are round and thick (ZPAL/AbIII/397, personal observation).

The distribution of certain characteristics according to the relative individual age or type assignment is presented in Figs. 4–7. All plots show bimodal distribution of the parameters, which are independent of estimated relative individual age of specimens. UOPB1165 (Fig. 2) specimen not fitting any of this types might be the representative of a different taxon, although it was the most incomplete specimen, which may affect the result of its description. Some features were not described for this specimen, i.e., borders and shape of the ossification centre or radial ridges character (Table 1). Estimated size of the complete specimen is small, but the ossification degree is high. Its possible assignment to species other than *M. krasiejowensis* would be difficult without other findings.

In Table 5, the results of conducted statistical test are presented –F and T or U, dependent on the data distribution. Considering described types as different groups, quantitative and qualitative data shows that they differ significantly ($\alpha = 0.05$).

### Micro/nanoscale

Two types can also be distinguished according to the micromorphology of the ornamentation ridges and bone structure in cross-section. Clavicles assigned to type 1 do not possess striations (or striations, if present, are barely visible and sparse) and possess a low number of small capillary foramina at the slopes of the ridges (less than 7 per $100\,\mu\text{m}^2$). Usually they also have less than one foramen per 1 mm of ridge length and no distinct bumps or roughness at the top of the ridge (Figs. 8 and 9, Table 2). In cross-section they possess growth marks in close proximity within poorly vascularised upper cortex (Fig. 8).

Clavicles assigned to type 2 possess striations on the ridges and a greater number of small foramina (more than 7 per $100\,\mu\text{m}^2$). Usually they also have more than one foramen per 1 mm of ridge length and distinct bumps and roughness at the top of the ridge (Figs. 8 and 9, Table 2). In cross-section they possess growth marks separated by well-vascularised zones (Fig. 9). This difference in histological patterns is analogous to different growth strategies described for the example of long bones (*Teschner, Sander & Konietzko-Meier, 2017*).

### Remarks on other dermal bones
#### Skulls

Bimodal differences were found also in skulls (Table 3), which have been divided in the Ts1 and Ts2 types. The main characteristic of ornamentation of ossifying centers resembles either Tc1 (large, hexagonal, sparse polygons, almost no multipolygons; six specimens; Ts1) or Tc2 (small, irregular and dense polygons with common multipolygons; seven specimens; Ts2). There is also a visible difference in the spatial distribution of polygonal and radial ornamentations between Ts1 and Ts2 (Fig. 10). In the first type, radial pattern covers large areas of the skulls roof in their both preorbital and postorbital (postfrontal, postorbital, supratemporal bones) parts, while in the second it occupies much smaller areas.

An important fact is that the skulls classified as Ts2 are relatively small (averaging 28 cm in length) in contrast to Ts1 skulls (averaging 35 cm in length). However, this was not

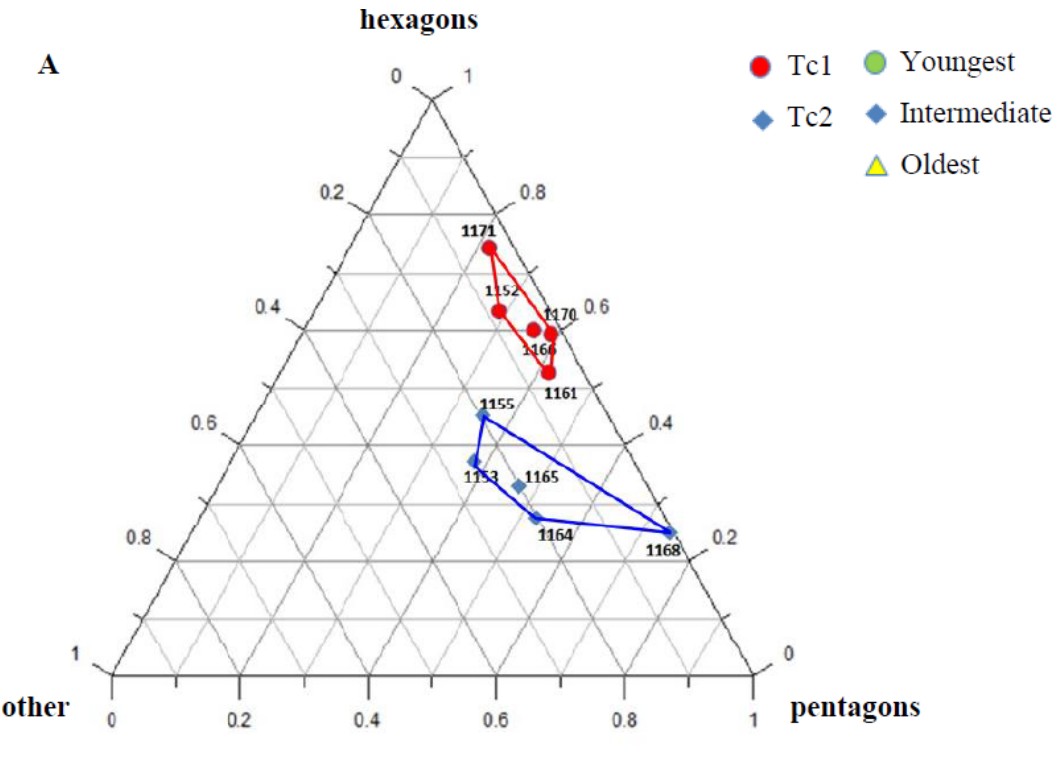

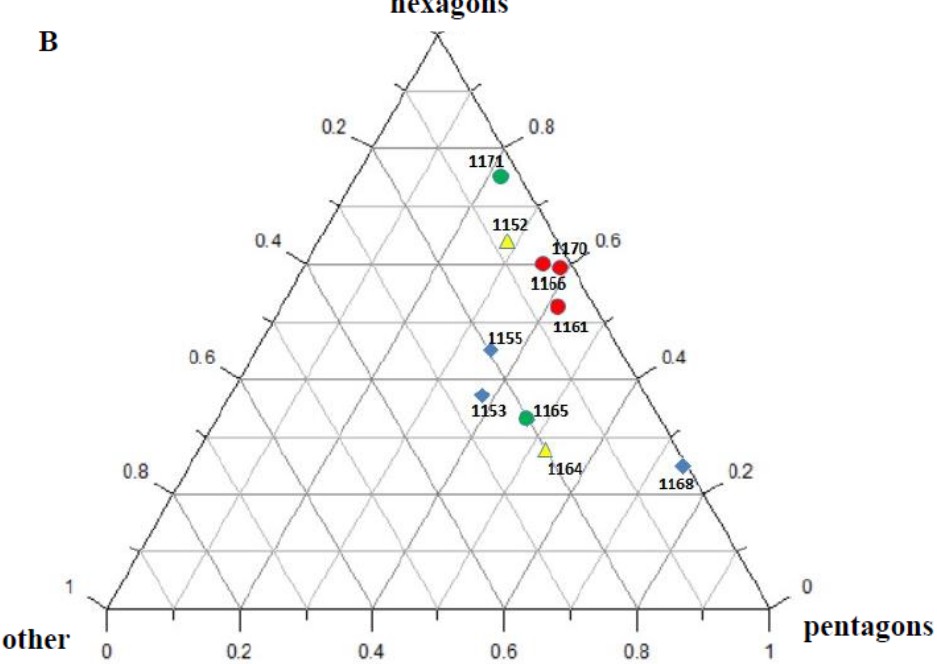

**Figure 7 Percentage of hexagons, pentagons and other polygons.** (A) Considering appointed types, showing two subsets within metoposaurid data. (B) Considering individual age, showing no subsets within metoposaurid data.

**Table 5 Statistical tests.** Bolded values are the end results (last stage of statistical testing) showing whether the samples are significantly different or not.

| | | p-value | |
| --- | --- | --- | --- |
| | | T1 | T2 |
| Av. polygon diameter/av. ridge width | Shapiro–Wilk Test | 0.55655 | 0.24746 |
| | Test F | 0.321792 | |
| | Test T | **0.00106** | |
| Multipolygon number | Shapiro–Wilk Test | 0.146977 | 0.04937 |
| | Test U | **0.001676** | |
| Ridge number/bone width | Shapiro–Wilk Test | 0.0703221 | 0.010253 |
| | Test U | **0.035556232** | |
| Qualitative data | Shapiro–Wilk Test | 2.587E-07 | 0.541135 |
| | Test U | **0.000194** | |

a rule. Among analysed skulls were two 35 cm in length (UO/PP04, 35 cm; UO/PP18, 35.4 cm) with different ornamentation types (Fig. 10, Table 3, Fig. 4).

## DISCUSSION

### Reasons for the observed variation in dermal bone ornamentation

The presented diversity in the dermal bone ornamentation of *M. krasiejowensis* may be the result of species diversity, ontogenetic diversity, sexual dimorphism and individual variation, different habitats of two populations or facultative neoteny.

1. **Species diversity.** Given that no differences were found in axial and appendicular skeleton characteristics—all analysis described metoposaurid material as one species, *M. krasiejowensis* (i.e., *Gadek, 2012*; *Konietzko-Meier & Klein, 2013*; *Konietzko-Meier & Sander, 2013*; *Teschner, Sander & Konietzko-Meier, 2017*) or in dermal bone measurements, it is also unlikely that the described differences in the analysed material represent differences between two species. Shape and ornamentation pattern of the clavicles (both described types) is strongly distinct from *M. algavrensis* or *Cyclotosaurus intermedius* (Figs. 4–6) (*Sulej & Majer, 2005*; *Brusatte et al., 2015*). Only the distinct character of the UOPB1165 specimen observed on the bivariate plots of countable features might suggests that this specimen does not belong to the same species. The occurrence of some other taxon is possible because of the redeposited character of the fossils. Moreover, in skulls (both types –Ts1 and Ts2), the prepineal part of the parietals is short and the expansion angle of the sutures separating the parietal from the supratemporal vary between 19 and 26° which is characteristic of *M. krasiejowensis* instead of *M. diagnosticus* (longer prepineal part and parietal expansion angle being around 13°) (*Sulej, 2002*) (Fig. 11). Also relatively narrow shape of the skulls and shape of the sutures (i.e., between frontals and narials or parietals) is typical of *M. krasiejowensis*, being distinct from *M. diagnosticus*, *M. algavrensis* (*Brusatte et al., 2015*) (Fig. 11) or *Cyclotosaurus* (ZPAL/AbIII/397). According to this all skull specimens belong to *M. krasiejowensis*. As only one species (*M. krasiejowensis*) can be described

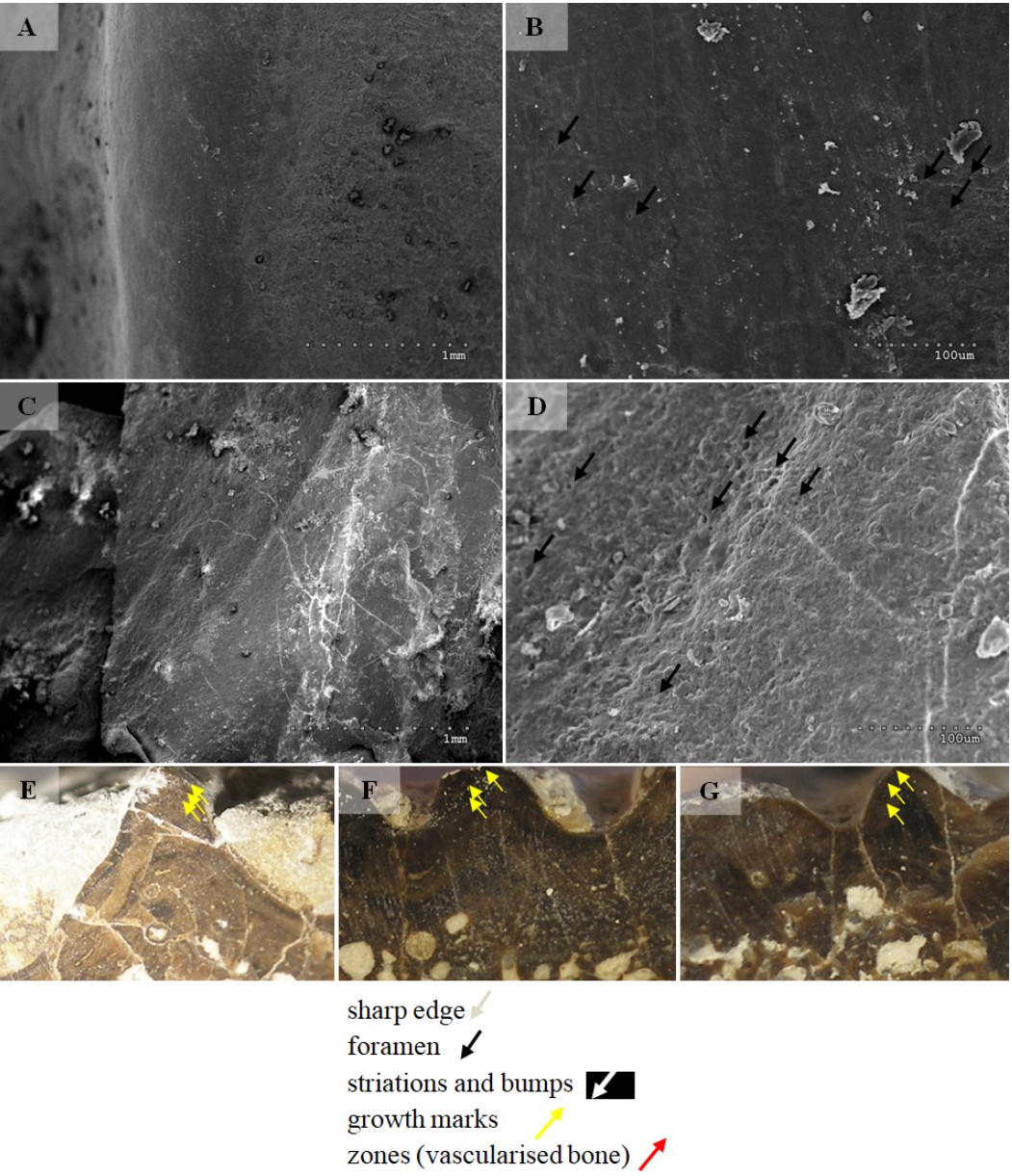

sharp edge ⬊
foramen ↙
striations and bumps ◥◣
growth marks ↗
zones (vascularised bone) ↗

**Figure 8** **SEM (SE) and histological observations of clavicle radial ridges for Tc1.** (A–B) UOPB1152. (C–D) UOPB1161. (E) UOPB1160. (F) UOPB1167. (G) UOPB1170.

considering skulls, and only this species was described in Krasiejów in over 15 years of studying metoposaurid material, it seems justified to consider all of the clavicles as belonging to *M. krasiejowensis*. With possibly one exception—UOPB1165.

2. **Ontogenetic diversity.** According to *Witzmann et al. (2010)*, all described specimens belongs to adult individuals, as they all can be assigned to the last stage of sculpture development (*Witzmann et al., 2010*: Fig. 6E). Although singular features may be connected with the age of the specimen, the method of determination of relative age for

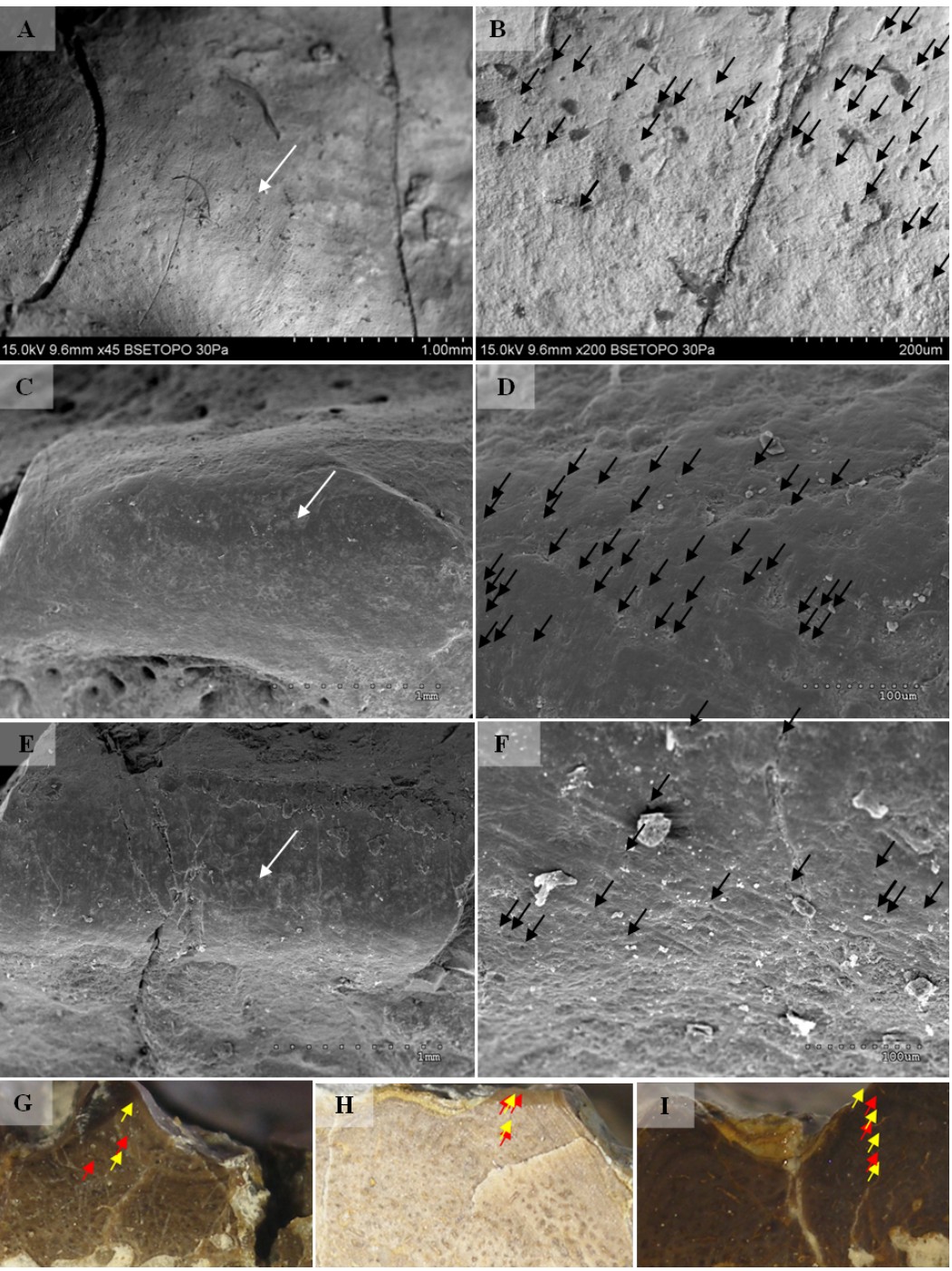

**Figure 9** **SEM (SE) and histological observations of clavicle radial ridges for Tc2.** (A–B) uam/kng/02. (C–D) UOBP1157. (E–F) UOPB1163. (G) UOPB1172. (H) UOPB1158. (I) UOPB1163.

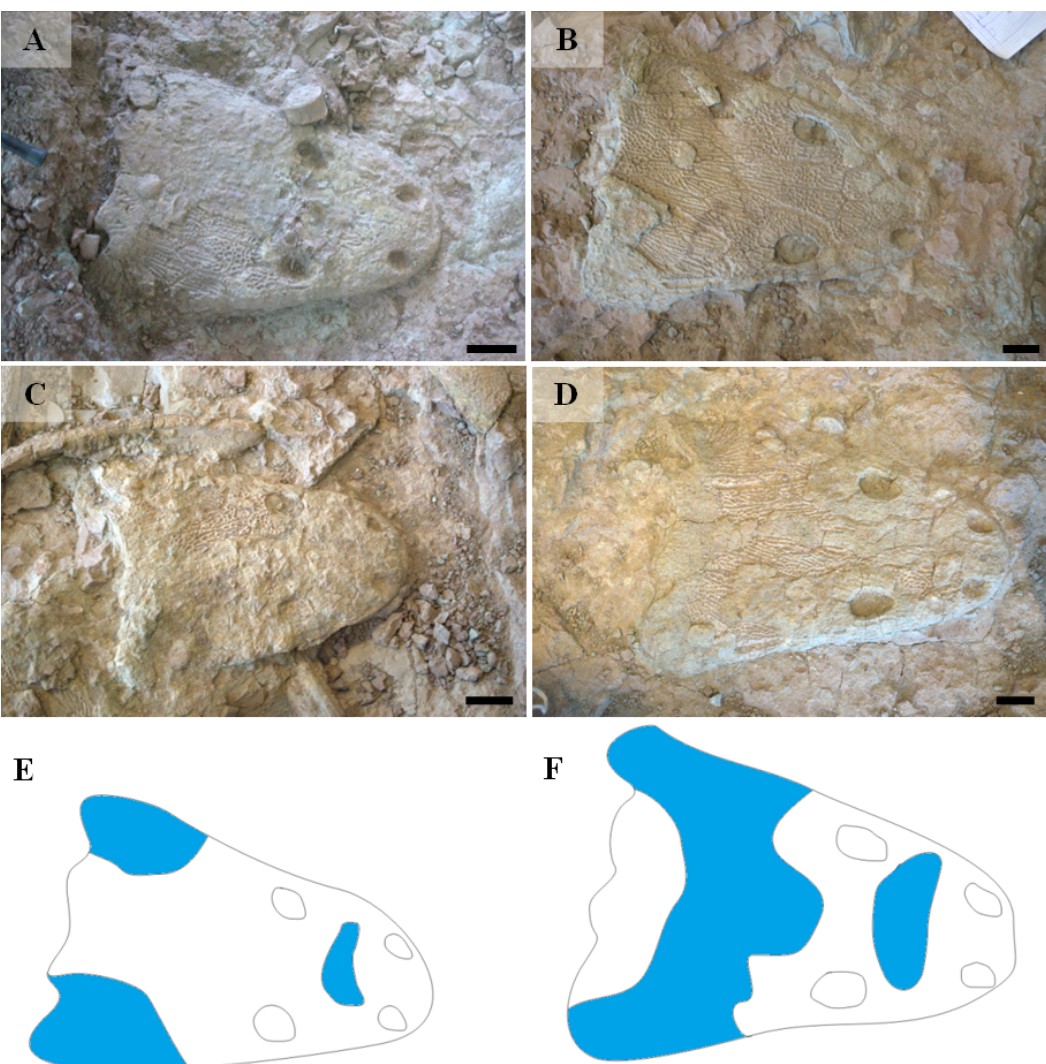

**Figure 10 Types of skulls ornamentation of metoposaurids from Krasiejów (*M. krasiejowensis*, as explained at Fig. 11).** Blue area represents surface covered with radial ornamentation. (A) UO/PP20. (B) UO/PP13. (C) UO/PP08. (D) UO/PP09. (E) Ts2 skull. (F) Ts1 skull. Scale bar 5 cm.

clavicles (youngest, intermediate, and oldest stages) based on the number of partition walls within the radial ornament shows that most of the analysed features, along with bone thickness, are not connected in this way. The youngest specimens possessed no partition walls between radial ridges. An intermediate stage was represented by specimens with developing partition walls within radial ornaments, and the oldest specimens possessed many well-developed partition walls between radial ridges. Additionally, clavicles described as the oldest stage, are the largest ones (UOPB1152 ~19,5 cm × 9,7 cm, UOPB1164 ~20 cm × 9 cm), while the youngest are usually of small size (UOPB1166 ~12 cm × 6 cm, UOPB1171 ~10 cm × 5 cm). Unfortunately the histology of dermal bones cannot be used to determine the exact individual age,

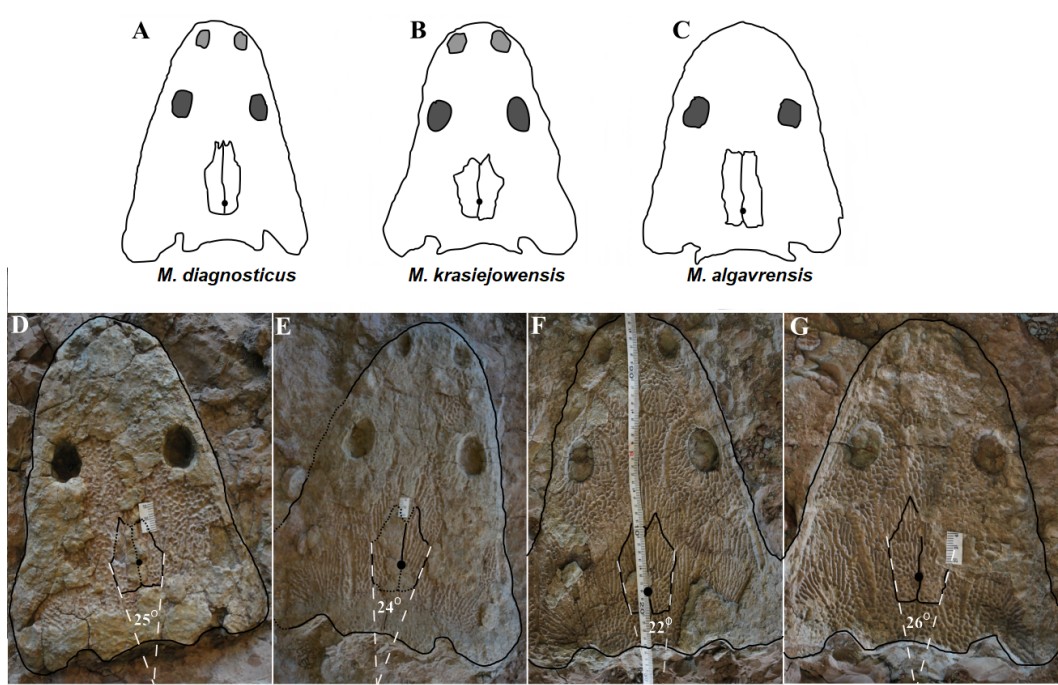

**Figure 11** Comparison of analyzed skull material (examples) with reconstructions of *Metoposaurus diagnosticus, M. krasiejowensis* and *M. algavrensis* skulls. (A) *M. algavrensis.* (B) *M. krasiejowensis.* (C) *M. algavrensis.* (D) UO/PP02 (Ts2). (E) UO/PP09 (Ts1). (F) UO/PP13 (Ts1). (G) UO/PP04 (Ts1). (A–C) after (*Brusatte et al., 2015*).

as different cross sections of the same bone reveals different stage of remodelling and counting the growth marks is unreliable (K Gruntmejer, pers. comm.; *Konietzko-Meier et al., 2018*; Figs. 8 and 9). The diversity of skull sizes assigned to different types also argues against ontogenetic diversity. Relatively small skulls possess more polygonal (adult; *Witzmann et al., 2010*) ornament than the largest skulls. In addition, there are no differences in the ratio of skull portions according to size; whereas in the metoposaurids, in the younger specimens, the orbits are placed further back on the skull relative to its length (*Davidow-Henry, 1989*), i.e., the area between orbits grew faster in temnospondyls than the orbits themselves. Polygon characteristics also indicate the adult stage in all skull specimens. *Rinehart et al. (2008)* and *Lucas et al. (2010)* also suggest that all individuals are adults. *Sulej (2002)* suggests that size of the clavicle depends on the age and recognized several clavicles of different size as an ontogenetic sequence. Nevertheless, this ontogeny cannot be used to explain ornamentation variety, as the two types of sculpture occur in both small and large specimens. The differentiation is also not the same as in the Rotten Hill, where age differences were proposed (*Lucas et al., 2016*). There are no size classes that can be correlated with sculpture variety in clavicles. In skulls, specimens assigned to type 2 are usually smaller, with exception of UO/PP18 (Table 4, Figs. 12 and 13).

3. **Sexual dimorphism.** In the described material there is lack of dimorphism in the shape of the skulls (*Urban & Berman, 2007*), clavicles or dentition (*Kupfer, 2007*). The

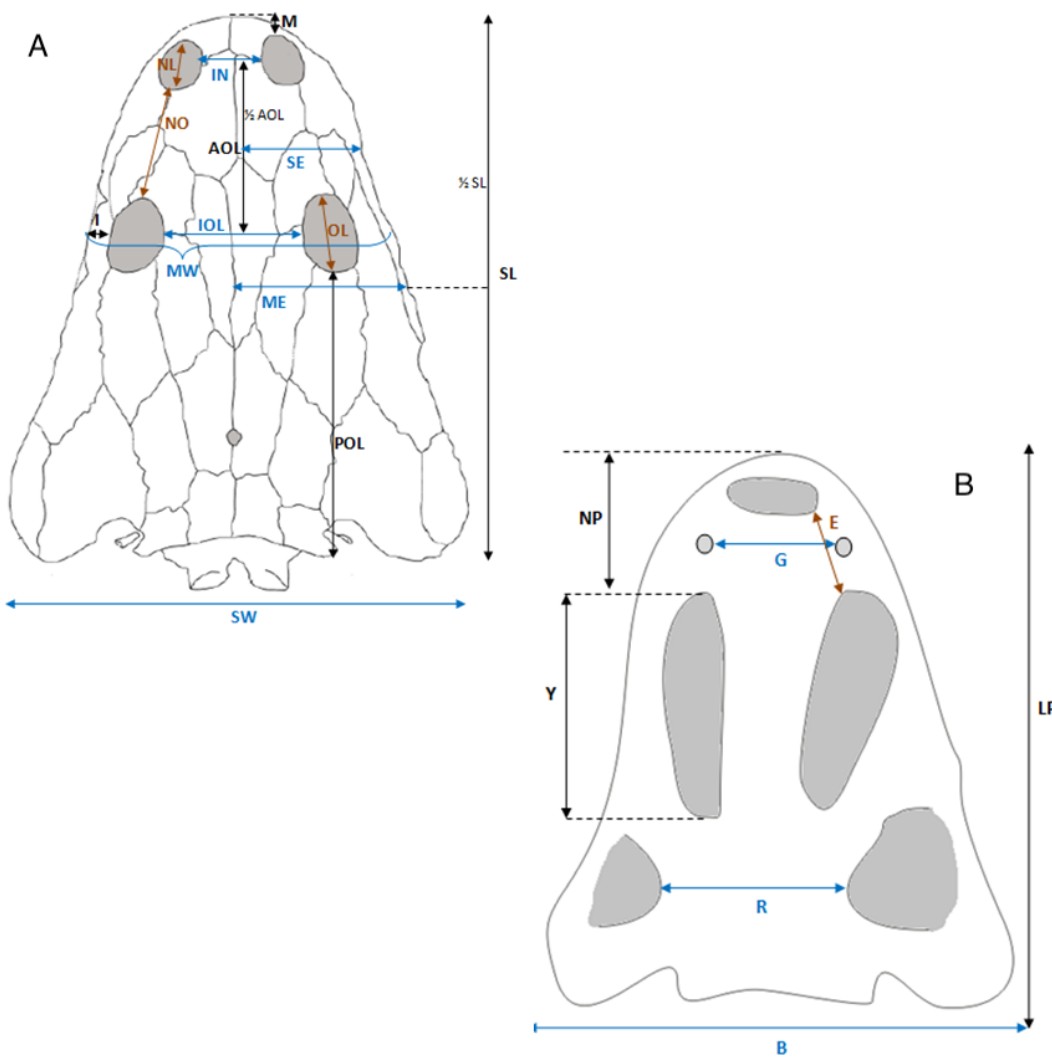

**Figure 12 Skull measurements.** (A) Skull roof. (B) Palate.

location of clavicles (under the skin and on the ventral side of the body) and discussed function of the ornamentation excludes its role as 'display structures' in mating rituals (*Kupfer, 2007*) in contrast to, i.e., *Zatrachys serratus* were spinescence and shape of the skull (rostrum) were considered as sexual dimorphism (*Urban & Berman, 2007*). Different growth strategy seen in clavicles (Figs. 8 and 9), skulls (K Gruntmejer, 2018, pers. comm.) and long bones (*Teschner, Sander & Konietzko-Meier, 2017*) ('seasonal' growth marks separated by vascularised zones or slower growth with growth marks in close proximity within poorly vascularised bone) rather do not indicate different sexes, but was ecologically controlled.

4.  **Individual variation.** The existence of two distinct ornamentation types with no intermediate patterns (Figs. 3–7) may support different ecological adaptations (see below) rather than individual variation as the only reason of diversity.

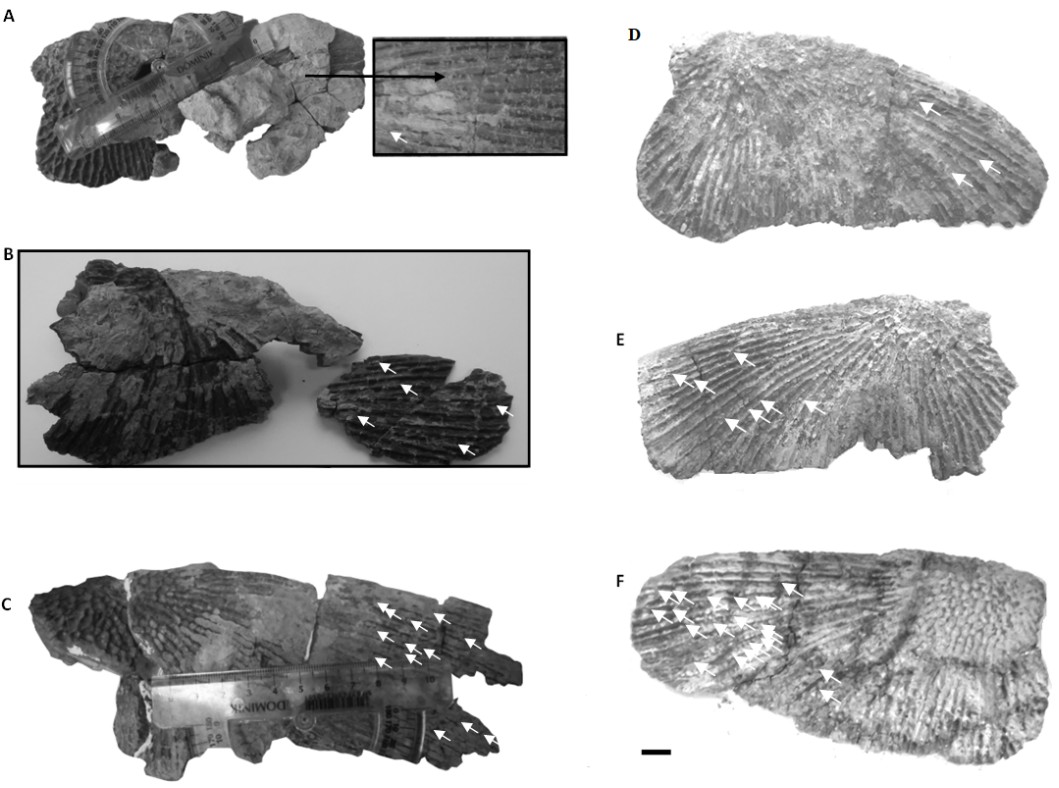

**Figure 13 Ontogenetic stages in clavicles.** Arrows show partition walls and ramifications within radial ridges. (A–C) Ornamentation type 2 (Tc2) from possibly the youngest specimen to the oldest one. (D–F) Ornamentation type 1 (Tc1) from possibly the youngest specimen to the oldest one. (A) UOPB1155. (B) UOPB1168. (C) UOPB1164. (D) UOPB1171. (E) UOPB1166. (F) UOPB1152.

5. **Different habitats**. Morphology of the dermal sculpture and vascularisation are not separable. Regularity of the ornamentation reflects the mode of life of temnospondyls to a certain degree. The coarser ornament, more pronounced ridges and irregularity is characteristic of rather terrestrial taxa (i.e., *Seymouria, Eryops*, see: *Witzmann et al., 2010*)—T2, while irregular sculpture represents rather aquatic animals (*Witzmann et al., 2010*)—T1. The variety seen within *M. krasiejowensis* allows expanding this conclusion, showing that the ecological difference (listed features) can be observed within one species. Metamorphosis is a hormonally induced and controlled process; thus, its results might be morphologically unequal even in closely-related taxa (*Fritzsch, 1990*; *Norris, 1999*) or within taxa (*Rafiński & Babik, 2000*; *Pogodzinski, Hermaniuk & Stepniak, 2015*). Because of this and the fact that amphibians, as animals very closely connected with the environment, are phenotypically plastic (examples below), the morphological diversity of the analysed material may be a result of differences between ecologically separated populations (geographic separation). Ecological separation of animals which remains are deposited in one bone-bed is possible, because of the bone-bed character (material partially redeposited, possibly from distant area, and partially local). Redeposition from different environments is suggested by the variant

infill succession in the pore system and trace elements contents in the individual remains (*Bodzioch & Kowal-Linka, 2012*; *Bodzioch, 2015*). The more aquatic population might have lived at a different site—fossils are redeposited and material might be transported even from Variscian Upland according to isotopic analysis of *Konieczna, Belka & Dopieralska (2015)*. Thus, geographical separation is a probable explanation, because the different ecological character of specimens might suggest that the two populations did not interbreed with each other. Time separation is also plausible. Some clavicles can be reworked more than once, being removed from older level than those which provided the skulls, which often seem to have a better preservation. The more terrestrial population probably lived at the site, where environment resembles modern Gilgai relief of Texas or Australia (*Szulc, Racki & Jewuła, 2015*) while more aquatic populations lived at some distance in larger reservoir(s). Although the presence of some large skulls with no abrasion or weathering does not support transport from a distant area, a brief transport however is plausible as the teeth in the mandibles and upper jaws are usually lost (*Lucas et al., 2010*). Other possibility is temporal diversity—gradually changing conditions of environment parallel with amphibian morphology/behaviour adaptation, however some intermediate ornamentation patterns should have been noticed in that case—see 'individual variation'.

6. **Facultative neoteny (paedomorphism).** Explanation assuming the same environmental differences between described morphotypes, but within a single population. The Late Triassic Krasiejów environmental conditions (dry and rainy season with possible periodic lack of food) may have even contributed to the formation of a neotenic population (*Duellman & Trueb, 1986*; *Safi et al., 2004*; *Frobisch & Schoch, 2009*). However, evidence of larval structures (i.e., branchial ossicles) in adult metoposaurids from Krasiejów is lacking. Nevertheless, facultative neoteny is possible (*Motyl, 2008*), as shown by the more radial (juvenile) sculpture on the large skulls of Ts1 (*Witzmann et al., 2010*). Facultative neoteny can be observed in several extant taxa, i.e., *Ambystoma talpoideum* with aquatic paedomorphic adults and terrestrial metamorphic adults (*Whiteman, Krenz & Semlitsch, 2005*). Breeding between such morphs is less common than within morphs, because paedomorphic adults begin to breed earlier (*Krenz & Sever, 1995*; *Whiteman & Semlitsch, 2005*). In this case *M. Krasiejowensis* Type 2 (Tc2, Ts2) reflects metamorphic adults that transform into somewhat terrestrial, while Type 1 (Tc1, Ts1) reflects (partially) paedomorphic aquatic adults. This is possible because larval development is dependent on the environmental conditions. In Late Triassic Krasiejów dry and rainy seasons occurred which is known thanks to the versicolor nature of claystone and faunal composition with, i.e., dipnoans (*Szulc, 2005*; *Szulc, 2007*; *Skrzycki, 2015*). Associated with these changes in water-level, food availability, living space, and competition (*Ghioca-Robrecht, Smith & Densmore, 2009*) may have influenced the preferred lifestyle. Metamorphosis into terrestrial or paedomorphic aquatic form is in this case the response to the individual expected success in the environment (*Wilbur & Collins, 1973*; *Whiteman, 1994*; *Michimae & Wakahara, 2002*) controlled by endocrine signals (*Pfennig, 1992*). Facultative neoteny in metoposaurids may occur in a single population (no geographical separation is needed) – spatial

separation of morphs may occur instead, with the paedomorphic concentrating in deeper habitats (*Whiteman & Semlitsch, 2005*).

### Ornamentation and lifestyle

The environmental differentiation is the most likely explanation regardless of whether caused by facultative neoteny or existence of two ecological types. Thus, described ornamentation types reflect more aquatic (Type 1) and more terrestrial (Type 2) morph of *M. krasiejowensis*. In modern limbless serpentine amphibians (Gymnophonia: Apoda) and lizard-like salamanders (Caudata: Urodela), larvae resemble miniature adult specimens. Metamorphosis is gradual and there is little reorganisation of body plan (*Zug, 1993*). In fossil amphibians, body plan reorganisation was also minimal and rather gradual (*Boy, 1974*; *Boy, 1988*; *Boy, 1990*; *Schoch, 2002*; *Schoch, 2004*), although its rate (trajectory: *Schoch, 2010*) might differ between taxa depending on their habitat (*Schoch, 2009*). This is also the cause that there are no other features suggesting more aquatic or more terrestrial lifestyle. Such changes, like differences in lateral line morphology, require more 'evolutionary effort', while changes in ornamentation are probably fast, reversible and do not require genetic changes (*Rafiński & Babik, 2000*; *Babik & Rafiński, 2000*).

Typically aquatic taxa are characterised by slow changes (low trajectory), sometimes with incomplete ossification of the pelvic region and limbs (last stages of ontogenetic trajectory). Terrestrial taxa are characterised by faster metamorphosis (high trajectory, with particular phases condensed within a short period of time), including final phases (limb ossification) enabling locomotion on land. The trajectory of semi-aquatic taxa lies between the two above-mentioned types.

This is an example of heterochrony. The length and composition of the temnospondyl ontogenetic trajectory is ecologically controlled (*Schoch, 2010*). Metamorphosis in this case might be described as extreme heterochrony, because many phases are condensed within a short time span (*Alberch, 1989*).

Ontogenetic trajectory and the morphology of adult specimens and their sizes may differ between various environments inhabited by representatives of the same taxon (*Schoch, 2010*). There are several examples of such diversity, such as differences observed in the length of the hind limbs of modern frogs (*Schmidt, 1938*; *Dubois, 1982*; *Emerson, 1986*; *Emerson, Travis & Blouin, 1988*; *Rafiński & Babik, 2000*) and the morphology of extinct temnospondyls: the ontogenetic rate and dentition of *Apaeton* (*Schoch, 1995*); the size of *Micromelerpeton* (*Boy & Sues, 2000*; *Schoch, 2010*); the morphology of *Sclerocephalus* (*Schoch, 2010*); the branchiosaurids (*Werneburg, 1991*; *Werneburg, 2002*; *Werneburg, Ronchi & Schneider, 2007*); and the plasticity of the plagiosaurid *Gerrothorax* (*Schoch & Witzmann, 2012*; *Sanchez & Schoch, 2013*). Polyphenism (environmentally controlled polymorphism) exists in a wide range of extant taxa (*Roff, 1996*) in adults (*Whiteman, Krenz & Semlitsch, 2005*) and tadpoles (*Collins & Cheek, 1983*; *Pfennig, 1990*; *Pfennig, 1992*; *Walls, Belanger & Blaustein, 1993*; *Nyman, Wilinson & Hutcherson, 1993*; *Michimae & Wakahara, 2002*; *Pfennig & McGee, 2010*).

Dimorphism in bone characteristics of metoposaurids from Krasiejów can be seen in dermal bones as well as in non-dermal skeletal elements from Krasiejów. Two types

connected with growth trajectory were seen in histological observations of metoposaur skulls (K Gruntmejer, 2018, pers. comm.), humeri (*Teschner, Sander & Konietzko-Meier, 2017*), and the morphology of femora (*Konietzko-Meier & Klein, 2013*).

New facts about metoposaurids from Krasiejów show that they were not fully aquatic animals. 3D computational biomechanics analysis of the skull of *Metoposaurus* show that it was capable of biting prey in the same manner as semi-aquatic and terrestrial animals like *Cyclotosaurus* or modern crocodiles (*Gruntmejer, Konietzko-Meier & Bodzioch, 2016*; *Fortuny, Marcé-Nogué & Konietzko-Meier, 2017*; *Konietzko-Meier et al., 2018*).

The described diversity is consistent with the experiment of *Schoch (1995)* and the results of *Werneburg (2002)* and *Schoch (2010)*. One of the *Metoposaurus* ornamentation types from Krasiejów (T2) thus represents a more terrestrial form (associated with the more variable and unstable environment of a river or a small lake or the metamorphic adult form of facultative neotenic population), while the other (T1) represents forms more closely related to water (a large lake habitat or partially paedomorphic aquatic adults) (ecological populations—as stated by *Witzmann et al. (2010)*; but described as species-specific; neoteny as described by *Whiteman, Krenz & Semlitsch (2005)*).

The adaptations in T2 favouring a more terrestrial lifestyle are:

a. The increased mechanical strength of the bones (*Rinehart & Lucas, 2013*) (coarser, denser, irregular sculpture, thicker clavicles);

b. Protection from a greater number of blood vessels, improving thermoregulation (*Gadek, 2012*) (denser sculpture, more numerous polygons and radial rows, more numerous microforamina);

c. Stronger integration of bone and skin, which is thicker in terrestrial amphibians and exfoliates (*Zug, 1993*; *Schoch, 2001*) (coarser, denser sculpture, microstriations);

d. Stronger connection of the pectoral girdle elements and, potentially, limbs (expanded anterior projection of the clavicle);

e. Faster growth revealed by histological structure (growth marks separated by zones of highly vascularised bone).

The more terrestrial character of one of the population may also be proved by:

f. Faster (at younger age) metamorphosis revealed by smaller skulls;

g. The length of limb bones not correlated with individual age (*Teschner, Sander & Konietzko-Meier, 2017*) or a slender or robust femur (*Konietzko-Meier & Klein, 2013*); 10% elongation of limbs in Anura distinctly increases migration capabilities (*Pogodzinski, Hermaniuk & Stepniak, 2015*; M Pogodziński, pers. comm.).

The dimorphic character of clavicles described herein and the two growth patterns of dermal and long bones (humeri) (*Teschner, Sander & Konietzko-Meier, 2017*) suggests that the ontogeny of specimens assigned to *Metoposaurus krasiejowensis* could have proceeded via a different growth rate and time span of metamorphosis, caused by differing environmental conditions. The similar number of specimens from both populations (Tc1/Tc2—44%/56% and Ts1/Ts2—53%/47%) suggests stable populations.

Apart from dermal bone ornamentation, the degree of ossification and variation in skull sizes divides metoposaurids into two groups. Smaller skulls occur in the more terrestrial type, as in *Micromelerpeton* from Germany, where smaller specimens represent an unstable

lake environment (*Boy & Sues, 2000*). The described type T2 reflects a more terrestrial or riparian habitat, where environmental conditions are variable and amphibians are forced to change their dwellings more often (migration between watercourses or 'stream-type' small, drying lakes; *Werneburg, Ronchi & Schneider, 2007*). It does not mean that 'more terrestrial/stream' metoposaurids moved efficiently on land. Modern salamanders can migrate between rivers and lakes by 'pond-hopping' (*Zug, 1993*). The first type reflects a more stable habitat, possibly a large lake, where animals are not forced to migrate ('pond-type'; *Werneburg, Ronchi & Schneider, 2007*).

Geological, sedimentological, and other analysis of the Krasiejów site shows that both of these habitats—episodic rivers and ponds at the excavation site and a large reservoir in close proximity—may have occurred there (redeposited charophytes and Unionidae bivalves; *Szulc, 2005*; *Szulc, 2007*), and that conditions changed over time (*Dzik & Sulej, 2007*; *Gruszka & Zieliński, 2008*; *Bodzioch & Kowal-Linka, 2012*). Differences in dermal bone ornamentation constitute an adaptive answer to changes in the environment (temperature, water level, food availability) over time or to geographical differentiation of habitats, i.e., faster metamorphosis (at smaller size) as an answer to higher temperatures; or metamorphosis into terrestrial adult vs. transformation into aquatic paedomorphic individuals.

Rapid changes in the ornamentation morphology in one population (or part of the population, when weather conditions favour such solution) are possible because they are the effects of hormonally induced metamorphosis. The water temperature in which larvae live strongly affects ectothermic animals. The growth of amphibians and larval development both depend on external environmental factors. At higher temperatures, not only metabolic rate but also development rate increases (*Motyl, 2008*). Low temperatures reduce development rates to a greater extent than they reduce growth rate, as a result of which amphibians metamorphose after achieving larger size (*Wilbur & Collins, 1973*) (Ts1 skulls are usually larger than Ts2 skulls). Prey abundance might exert some influence as well (*Motyl, 2008*), but probably not as much (*Blouin & Loeb, 1990*).

The Krasiejów ecosystem changed over time. The Late Triassic climate favoured evolution of freshwater environments. In Krasiejów, small periodic reservoirs, probably also inhabited (as in the environments of the Saar-Nahe Basin), occurred along with larger more stable ones (*Szulc, 2005*; *Szulc, 2007*; *Gruszka & Zieliński, 2008*; *Szulc, Racki & Jewuła, 2015*). Small reservoirs (and potentially with higher temperature) or periodic rivers forced earlier metamorphosis, dwelling on land, or migration between lakes and watercourses. On the other hand, larger lakes or the proximity of a large reservoir enabled the development of a fully aquatic (*Szulc, 2005*), possible neotenic population.

Large reservoirs, stable over long periods of time, enable the development of a fully aquatic (neotenic?) ecotype T1 (Tc1, Ts1), reducing the need to dwell on land by virtue of providing:

- enough room for numerous large specimens;
- shelter from mainland carnivores;
- stable, invariable conditions;
- potential lower temperatures.

The ontogenetic trajectories of the two metoposaurid ecotypes from Krasiejów cannot differ on a large scale, because they are assigned to the same semi-aquatic species. However, between types there was clearly some deflection into a more aquatic or more terrestrial form. In the case of a more terrestrial (stream-type) ecomorph, the trajectory would be more condensed (*Schoch, 2001*).

According to the described observations, it is possible to introduce an argument about the function of temnospondyl ornamentation into the discussion. There are several hypotheses as to the function of the ornamentation, which may have been:

1. mechanical strengthening of the bone (*Coldiron, 1974*; *Rinehart & Lucas, 2013*);
2. water-loss reduction (*Seibert, Lillywhite & Wassersug, 1974*);
3. integration of the bone and skin (*Romer, 1947*; *Bossy & Milner, 1998*);
4. improvement of dermal respiration (*Bystrow, 1947*);
5. thermoregulation (*Seidel, 1979*; *Grigg & Seebacher, 2001*);
6. acting as a metamorphosis marker (*Boy & Sues, 2000*);
7. buffering of acidosis and lactic acid build-up in tissues due to anaerobic activity (*Janis et al., 2012*).

The microstructural observations described in this manuscript support two hypotheses. Ornamentation increases the surface area of the bone (*Rinehart & Lucas, 2013*) and thus improves its thermoregulatory abilities and probably its integration with the skin, as histological thin sections show many Sharpey's fibres residing deep in the ridges (*Gadek, 2012*). Moreover SEM photographs presented herein show more or less numerous striations (skin and bone contact) and vascular foramina.

The hypothesis put forward by *Janis et al. (2012)* of dermal bone ornamentation developed in primitive tetrapods for the purpose of buffering acidosis and lactic acid build-up in their tissues due to anaerobic activity is also plausible. This would enable the amphibians to spend longer times on land and thus better exploit the terrestrial environment. This statement is in agreement with a study by *Witzmann et al. (2010)*, who stated that terrestrial forms (according to species or population) show more pronounced sculpture than aquatic forms.

## SUMMARY

The diversity of metoposaurid material from the 'Trias' site at Krasiejów (SW Poland) includes the character of ornamentation of clavicles and remarks of the ornamentation of skulls (although histological character suggests that all types of bones possess two types of bone growth). Similar differences in dermal bone ornamentation in Temnospondyli were cited as ecologically dependent by *Witzmann et al. (2010)*; however, these differences were assigned to particular taxa. Detailed analyses of large probe from one species shows that ecologically induced ornamentation differences can be observed within one species (from a single site).

Except for UOPB1165 specimen the taxonomical variety of the material was excluded. Observed differences in polygon shape, area, sculpture density, regularity and others (Tables 1 and 6) could be the result of individual, ontogenetic, sexual or ecological

**Table 6  Diagnosis and remarks on two populations of *M. krasiejowensis*.**

| | Type 1 | Type 2 |
|---|---|---|
| **Diagnosis–clavicle ornamentation** | Less numerous radial ridges | More numerous radial ridges |
| | Smaller ossification degree | Higher ossification degree |
| | Regular and fine ornamentation | Irregular and coarse ornamentation |
| | Sparse ornamentation | Gęsta ornamentacja kości skórnych |
| | Mostly hexagonal polygons | Mostly pentagonal (and other) polygons |
| | Few multipolygons | Numerous multipolygons |
| | Distinct border of ossification centre, square ossification centre | Border of ossification centre difficult to distinct, elongated ossification centre |
| | Polygonal ornamentation covering smaller area | Polygonal ornamentation covering larger area |
| | Less numerous microforamina and striations on the radial ridges | More numerous microforamina and striations on the radial bridges |
| | Growth Marks in close proximity within al most avascular upper cortex | Growth Marks separated by vascularised zones |
| **Remarks** | Mostly radial ornamentation in the postorbital part of the skull | Mostly polygonal ornamentation in the postorbital part of the skull |
| | Larger skulls | Smaller skulls |
| | Two growth patterns seen in femora and humeri (*Konietzko-Meier & Klein, 2013*, *Teschner, Sander & Konietzko-Meier, 2017*) | |

variation. Although some sort of sexual dimorphism or ontogenetic changes cannot be excluded, the most probable explanation for the described variation is ecological difference between two populations as stated by *Witzmann et al. (2010)*; or existence of facultative neotenic population. Described ornamentation types within one semi-aquatic species possess characteristic of either more-terrestrial or more-aquatic taxa. Some ontogenetic differences can be observed in both populations but they can be described separatedly in both populations with the smallest (youngest) specimens having low number of ramifications and partition walls within radial ornamentation and the largest (oldest) having high number of ramifications and partition walls.

Assuming that the more-terrestrial or 'stream-type' form can be distinguished by its smaller size (earlier metamorphosis), coarser and more complicated sculpture, more numerous ridges for protection of more numerous blood vessels, and a stronger connection between bones and skin for increasing mechanical strength for land-dwelling the more-aquatic or 'pond-type' form is characterised by greater size (later metamorphosis) and sparser, more regular ornamentation. Comparable differences in ontogenetic trajectories were described in *Sclerocephalus* by *Schoch (2010)*.

This ecological diversity corresponds with the geological description of Triassic Krasiejów, which includes redeposited material after flash floods, an environment with periodic rivers and ponds, and a large, more stable reservoir in close proximity, as described by *Szulc (2005)*; *Szulc (2007)*, *Gruszka & Zieliński (2008)*, *Bodzioch & Kowal-Linka (2012)*, and *Szulc, Racki & Jewuła (2015)*. The palaeoenvironment of the site, similar to modern Gilgai relief (*Szulc, 2005*; *Szulc, 2007*; *Szulc, Racki & Jewuła, 2015*) could be the habitat of more terrestrial population, while the more aquatic one could have lived at some

distance (closer—*Lucas et al., 2010*; or further—*Konieczna, Belka & Dopieralska, 2015*). One population with aquatic (paedomorphic) and terrestrial (metamorphic) individuals is also possible. In this case all metoposaurids could have lived in the same area with the paedomorphic concentrating in deeper habitats (*Whiteman & Semlitsch, 2005*) and metamorphic being more terrestrial (moving between shallow ponds and streams). Time difference between populations is also plausible.

The isotopic (or REE) analysis in the future may confirm the most probable explanation for metoposaurid ornamentation diversity and will provide valuable insight into the mechanism between it. More information about possible ornamentation character diversity can be obtained in the future considering distribution of shape (geometric morphometrics), possibly in all of the Metoposauridae.

## ACKNOWLEDGEMENTS

We wish to express sincere thanks to the reviewers: Spencer Lucas, Michael Buchwitz, the third anonymous reviewer, and the editor Graciela Piñeiro, who kindly improved language, added many important remarks and comments to an earlier versions of the typescript. We also would like to thank Tomasz Sulej for sharing access to the *Cyclotosaurus* material from the Museum of Evolution in Warsaw and Maciej Ruciński for his help in skull measurements.

### Funding
The authors received no funding for this work.

### Competing Interests
The authors declare there are no competing interests.

### Author Contributions
- Mateusz Antczak conceived and designed the experiments, performed the experiments, analyzed the data, prepared figures and/or tables, authored or reviewed drafts of the paper, approved the final draft.
- Adam Bodzioch conceived and designed the experiments, contributed reagents/materials/analysis tools, prepared figures and/or tables, authored or reviewed drafts of the paper, approved the final draft.

### Data Availability
The data are included in the Tables and Figures.

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
