# Peer review of "Ornamentation of dermal bones of Metoposaurus krasiejowensis and its ecological implications"

_PeerJ, doi:10.7717/peerj.5267_

## Round 0.1 · original submission · Major Revisions

Dear Mateusz Antczak and Adam Bodzioch,

As you can see, we have three review reports for your manuscript “Ornamentation of dermal bones of Metoposaurus krasiejowensis and its ecological implications” and unfortunately all of them manifested serious problems, mainly concerning the followed methodology and the lack of support that it gives to the conclusions to which you have arrived. I concur with Reviewer 2 in that the Ms. is a little disordered, and it is hard to understand the subject when there is not a complete development of the issue before presenting the conclusions. Perhaps you need to delimitate the nature and the identity of all the analyzed materials offering more photographs and interpretive drawings of all of them (be cautious about Reviewer 3 concerns on this, he explained clearly what you need to fix; also, the correspondence between isolated bones should be much better supported, as well as the characters from which you believe that all the analyzed bones belong to Metoposaurus). Accordingly, the taphonomic circumstances that constrained the preservation of the supposedly related bones should be addressed for all the selected materials. Then, you can describe the methodology that you selected to identify possible differences in ornamentation of dermal bones taking into account the recommendations of the three reviewers about this matter and choose which you think will reflect better the goal of your contribution. Finally, you can present a discussion about the significance of your findings.

Taking into account all the above mentioned troubles found in your manuscript, it must be provisionally accepted for publication under Major Revisions.

Please, pay carefully attention to all the constructive recommendations that you have received from the reviewers and resubmit your manuscript. We hope to see your improved contribution soon.

Best regards,
Graciela Piñeiro

Reviewer 1 ·

Basic reporting

This manuscript unfortunately appears to overlook much of the modern literature on both modern and fossil amphibians. This is reflected in various easily-remedied nomenclatural errors throughout the text as well as more serious errors in amphibian evolution and osteology. In particular, the authors appear to ignore substantial literature on modern amphibian morphology, osteology, development, polyphenism, and histology. As a result, the context used to interpret the variation observed here is generally not consistent with the current state of the field. In some cases, the criteria used to distinguish between interpretations do not have any basis in the literature or in amphibian biology more generally. A discussion of polyphenism without any reference at all to the extensive work on polyphenism in the mole salamander complex Ambystoma, or in Hynobius, or even in classic anuran systems such as Hyla or Pelobates, seems to be a major oversight. Similarly, there is an extensive literature on dermal ossifications in amphibians, reptiles, and fossil early tetrapods, and that has not been referenced here either.

Experimental design

The data which are the basis for this study are categorical scores for a number of variables. In a few cases, measurements are taken. A series of biplots are presented and then a principle components analysis is used to combine these data into a single ordination space. I am concerned about several things: the independence of different metrics measured here, the appropriateness of the methods used for analysis, and the manner in which these results and subsequently interpreted. The data as collected here are themselves highly nonindependent. For example, the authors treat “average polygon diameter/avg ridge width,” “polygon size,” and “polygon shape” as different independent metrics. The first two metrics are obviously not independent at all, and the shape of a polygon will likely affect the diameter, depending on how diameter is measured (i.e. an oblong polygon may have a different diameter than an equilateral one). As a result, some of the comparisons themselves were essentially meaningless (e.g. Figure 7, where percents of hexagonal and pentagonal polygons are plotted against each other, producing a line that corresponds to y=1-x. This is essentially tautological: %hexagonal and % pentagonal polygons together should add up to approximately 100%, minus a handful of polygons that do not fit these categories. Furthermore, some of these measures are means but the errors are not reported and do not appear to be incorporated into the analysis. This has a very real potential to bias the results.
This nonindependence of measurements makes it difficult to interpret the ordination space produced in the PCA. PCA is not an ideal metric for categorical data; PCoA or NMMDS might be a more appropriate ordination method. I would also prefer to see ordinations of the sculpture data and the cranial morphometric data done separately in order to compare the clustering achieved from skeletal morphometrics to the clustering achieved from the sculpture data. That, to me, would be more convincing than the approach taken here. Finally, when conducting a PCA OR PCoA, your number of samples should exceed the degrees of freedom of the underlying dataset. This is a rule that many workers ignore, but this limits the inferential power of the PCA as conducted here.
Another thing to think about is that much of the metrics taken here are mean measurements of diameter and so on. Distribution of shape may be a more important factor here than mean shape. In my initial skim of the paper, I thought this was actually what had been undertaken, but was disappointed to find that it was not. Something like a semi-landmark approach where each cell is traced and registered so that the complete interclavicle is represented by a distribution of points in morphospace rather than a single mean point. These distributions could then be compared in a more meaningful manner than simple means. In other words, I think the underlying question could be quite interesting, but the execution here is lacking.

Validity of the findings

Given my criticisms of the methods and context, I am not confident that the findings are valid. The statistical treatments are not conclusive. However, the possible hypotheses these methods are supposed to differentiate between are never clearly stated and the basis for accepting or rejecting alternative hypotheses are not sufficiently supported or justified with reference to the literature. Some of this could be addressed by moving parts of the discussion into the introduction, but some of the inferential work is not actually consistent with the current body of knowledge in either modern amphibians or their fossil relatives.

Additional comments

I have read the manuscript “Ornamentation of dermal bones of Metoposaurus krasiejowensis and its ecological implications.” The authors conduct a multivariate analysis of bone sculpture of an assemblage of temnospondyls from the Triassic of Poland, finding two morphotypes. They then explore possible ecological implications of these two morphotypes, suggesting that these may represent an obligate aquatic population and an obligate terrestrial population living in the same area. A number of workers have speculated on the importance of bone sculpture both for the physiology of early tetrapods as well as for taxonomy, so a detailed study of this would be a welcome addition to the literature. However, I am skeptical of the methodology used in this particular study. I am also not satisfied that the authors provide sufficient and accurate context from both fossil and modern amphibians for their interpretations. As a result, I cannot recommend publication of this manuscript at this time.

I am sorry that this review is not more positive. I think the underlying question here (i.e. what does dermal bone sculpturing in temnospondyls represent?) is an interesting one. I am not sure a collection with a sample size of n=18 is the best test case, but I can imagine some ways to examine this in more detail even with the current taxonomic sample, including methods that would not require a substantial amount of additional legwork so long as all specimens are photographed in standard positions. However, drawing conclusions will be difficult with a small dataset such as this, and requires dealing directly with a lot of the modern literature on amphibian skeletal biology, which the authors do not do. I also think the authors need to end their interpretations earlier, at the presence of two skeletal morphs (if that conclusion is upheld) rather than drawing interpretations about different terrestrial and aquatic lineages with all of the detailed speculation following that inference. Much of the biological and evolutionary interpretation appears to be derived from first principles rather than reference to actual biology of living or extinct animals, which leads me to question whether the criteria proposed by the authors have any basis at all.

Again, I apologize that this is not a more positive review. I hope the authors will be able to take these comments in stride and restructure this study in a manner which is more likely to produce interpretable results. I do think the authors have an interesting question, but I think that a more sophisticated approach to both data collection and analysis as well as the biological context are necessary before this is publishable.

·

Basic reporting

The data presented in this article are important and should be published. However, the current presentation is incoherent and somewhat disorganized, so it is difficult to understand what these data about metoposaur ornamentation may be telling us—indeed, I am not sure the authors of the paper are certain of that. So, there needs to be a substantial rewrite and reorganization of this article. I suggest the following: (1) present the ornamentation data as is mostly done already, but eliminate the PCI as it is unnecessary and shows nothing that bivariate plots or histograms cannot show; (2) analyze the distribution of the ornamentation types and other parameters of the Krasiejow metoposaurs—in other words, make it very clear what correlates with what in these metoposaurs; (3) present a clear discussion of what the possible significance of these distributional data is; (4) choose the most likely conclusion about the significance.

Spencer G. Lucas

My more specific suggestiosn are keyed to numbers on the pdf:

1. Also see Lucas 2015 in Ann Soc Geol Pol.
2. I think the PCI is not needed. It tells us nothing that cannot be more easily obtained from simple plotting.
3. What about the two subspecies of Metoposaurus from Krasiejow named by Sulej? What is their relationship to this?
4. I don’t understand the possible separation, given that this is one metoposaur bonebed that is being analyzed? What is the basis for either geographic or stratigraphic separation in the sample?
5. There is no coherent argument here.
6. What is “T2”?
7. Barite in pores? Why isn’t this simply a function of diagenesis or post-diagenesis?
8. Is there a correlation between dermal bone structure and limb bone length?
9. Not a metoposaur.
10. Explain how?
11. So, there is a hypothetical Late Triassic waterbody somewhere else that the actual Krasiejow bonebed? How can we know what amphibians lived there?
12. These comments on isolated skulls are anecdotal at best. What about variation in the large sample of skulls of Koskinonodon described by Lucas et al. 2016 (NMMNH Bulletin)?
13. I don’t follow this conclusion—it was not developed in previous text.

Experimental design

No comment

Validity of the findings

See earlier comments

Additional comments

See earlier comments

·

Basic reporting

The manuscript entitled "Ornamentation of dermal bones of Metoposaurus krasiejowensis and its ecological implications" by Mateusz Antczak and Adam Bodzioch is fairly well written and organized but has a number of shortcomings - see below - that I think need to be addressed before re-submission. The English language appears to be sufficient (with minor errors) and the necessary literature is well covered

(1) Presentation of data: If the manuscript is submitted as a Latex file, all tables should be included by means of Latex formatting styles and not as badly resolved raster images. Additionally all data tables should be included as separate data files (txt, xls or else) – so that they can be readily accessed by the reviewers and other researchers.

(2) Inclusion of photos/interpretation drawings for all included dermal bones: In order to reproduce the measurements and character codings presented in the data tables 1-5 it would be helpful, if not necessary, to illustrate all of the ornamentation patterns that form the basis of these datasets - preferably for each specimen with inclusion of (a) a photo and (b) a vector-graphic-like interpretation drawing of the ornamentation patterns (as figure or supplemental file).
* * *
Experimental design

(3) The derivation of two morphotypes from the provided dataset is not fully clear and appears to be based on certain groupings/clusters visible in certain variable plots and principal component plots - plus the arbitrary decision that some of the data points shall belong to group A and the others to group B. The authors should decide _before_ doing any of the analyses whether

(a) they have no clear a priori way of distinction between different morphologies and now want to find out via some method of (unsupervised) classification based on their newly raised data if there are any clearly separable groups or

(b) they follow the idea that there is a clearly visible (qualitative) difference between two or more morphologies which is taken as given and shall be tested by means of multivariate analysis based on morphometric data

If they choose (a) cluster analyses (with varying linkage algorithms and distance measures) could be the method of choice to have an unbiased way of distinction between groups; if the authors choose (b) they could use e.g. PCAs/bivariate plots for illustrating differences between predefined groups (what they already did) and (multivariate) analyses of variance for testing. In the present version of the manuscript the reasoning behind doing the multivariate analyses remains fuzzy. (Is a quantitative distinction of morphotypes necessary if I already have got qualitative distinction criteria - for what reason?)

(4) The authors include the different types of dermal bones – skull-roofing bones, clavicles and interclavicles – and assume a correspondence between similar (?) ornamentation morphotypes I and II in skull bones and clavicles. However they do not test this assumed correspondence quantitatively (based on ornamentation characters/measures) nor with the help of associated remains/complete Metoposaurus specimens that include both, skull and pectoral girdle which would offer additional proof that skull and interclavicle ornamentation types actually correspond (belong to the same type I or type II individual).

The authors cite other studies by Teschner and Konietzko-Meier (2015), Konietzko-Meier and Klein (2013), Bodzioch and Kowal-Linka (2012), Bodzioch (2015) that show "dimorphism" in other bone characteristics, e.g. of femora and humeri, and they think that this kind of morphological variance is reflected by the signal they have found in dermal bones - but there is the similar problem as for other ornamented dermal bones that in case of isolated bones/ incomplete specimens no one can really tell which femora and humeri belong to which kind of clavicles.
* * *
Validity of the findings

(5) While the idea of two separable morphotypes appears to be supported by the data (despite the small sample size, but see also methodological problems described above), the authors’ interpretation of these types as ecomorphotypes, i.e. caused by ecological differences, appears to be rather vague for several reasons
- vague definition of ontogenetic stages based on a study by Zalecka (2012) - better would be other criteria for individual age, such as body size proxies or histological features of the dermal bones (e.g. growth marks, degree of remodelling)
- vague idea that a certain type of polygonal ornamentation is indicative for adult specimens and more ridge-like patterns for juveniles - is there a sufficient ontogenetic series to prove that relationship for M. krasiejowensis?
- lacking analoga of ecomorphotypes with comparable distinct dermal bone ornamentations in other temnospondyl species (the authors have cited examples for ecomorphotypes in other species but not for ecology-based intraspecific variation in ornamentation patterns)
- lack of complete specimens that could prove that certain types of clavicles actually corresponds to morphotypes of other bones (skulls, humeri, femora...)
- lacking evidence that the clavicle morphotypes I and II actually come from different environments and were deposited together or other (e.g. isotopic) data that could prove distinct ecologies for the individuals to which the distinct clavicles types belonged.

Thus I would refrain from calling the described morphotypes "ecomorphotypes" and I would not consider the author's interpretation of these types as intraspecific variation due to different ecologies as yet well supported by the data at hand. Arguably the discussion and conclusion should reflect somewhat better the remaining uncertainty about the meaning of described morphotypes.
* * *
Additional comments

(6) Diagrams: Better leave out regression functions completely - never plot them for mixed samples. They make only sense for presumably homogeneous subgroups.

(7) The subheadings "Diagnosis: clavicles", "Remarks on [other?] dermal bones" , "Possible solutions" sound strange/awkward - can't you find better ones? "Ornamentation and lifestyle" is not far away from "Ornament function" in terms of content, because function and lifestyle are closely related - why not discussing these points in one concise chapter of the Discussion section to avoid redundancy/overlap?

(8) I added a number comments to the manuscript pdf - suggestions for improvement and sometimes questions about points that remained unclear.

The points discussed above are in accordance with a major revision.

Sincerely yours

Michael Buchwitz, 14th December 2017

---

## Round 0.2 · Minor Revisions

Dear authors,

I have received the new review reports about your reviewed manuscript. As you will see, both reviewers acknowledged that the changes that you made into the former manuscript improved it very much; however, more work is still needed. Although their requests for more improvements of the manuscript can be considered as minor revisions in the general context, they are extremely important for the impact of your research. Thus I hope you will make the efforts needed to get an acceptable version very soon.

I strongly agree with the reviewers’ comments and recommendations, in particular with two or three aspects that they remarked:

1- A deep and detailed taphonomic section should be included. For instance, you have to take into account time averaging for the material that you find reworked and try to ensemble that information to the sedimentological framework known for the bone-bearing unit. It is okay that you cite people that have worked on the geology of the deposits, but in this case, it is necessary that you mention in your manuscript those aspects that help you to better support your hypotheses about the existence of two populations, maybe separated by time. Even you have to mention that there is other temnospondyl in the Krasiejów formation, particularly because both are different in the skull morphology but not so concerning the clavicles. Maybe you can include a more detailed description of the taxa found into the bone bed from which the clavicles and the skulls were collected; all these information can help the reader to have a more close view of the manuscript subject. See below also, the comments that I wrote after the first read of your manuscript, and that I have adapted to the new version after the last reviews. I think that they would be helpful for improve this and other aspects.

2- As you yet fail to demonstrate the significance (if ecological, ontogenetic or even taxonomic) of the two morphotypes certainly found, maybe you can modify your text leaving the corresponding uncertainties for each of the analyzed hypotheses. Otherwise, you could focalize onto the skulls, provide taphonomic information like if they were found associated in the same level, or they come from different excavation sites of the formation. Also, you have to say how you know that the materials that you used belong to a single taxon, I don’t know if you controlled that. There are at least three species of Metoposarus whose lacrimals are very similar in morphology. For Metoposaurus diagnosticus krasiejowensis the prepineal area is very short (after Sulej, 2002), but it is really hard to determine the position of the pineal foramen from the specimens that you figured, Maybe you could provide better images or make interpretive drawings (schematic ones would be enough) to demonstrate at least the presence of this character. For the clavicles there is not even a single feature that could help you in the diagnosis, so, how you know that these clavicles belong to Metoposaurus diagnosticus krasiejowensis?

3- Also, it is important that you clarify if the compared ornamentation pattern was observed at the same region of the skull (for instance, the amount of radial ridges in the ante -orbital area of all the analyzed skulls). As you should know areas of fast growing are characterized by long grooves (after Sulej, 2007, paper that should be included and commented, see below). That was not clear to me from the reading of the manuscript.
Sulej, T. 2007. Osteology, variability, and evolution of Metoposaurus, a temnospondyl from the Late Triassic of Poland. Palaeontologia Polonica 64:29-139.

I think that if you can address the mentioned weaknesses of your manuscript you could have a very interesting subject here.

Following, I am attaching some of my primary observations that were adapted to the current version of the manuscript.

Figure 10 where skulls are shown must have the scale bar included into the figure, please.
Line 15. Remove “amphibians”
Line 59. Describe please also the upper fossiliferous horizon.
Lines 64-67. It is not clear here from which horizon the analyzed materials come, and if the clavicles were reworked from underlying levels, as they don’t seem to have been collected in situ as the skulls. Also, one skull is housed in a scientific collection, and what happened to the others? Please, clarify.
Line 89. You should clarify that the SEM analysis was performed on clavicle ornamentation.
Lines 91-96. You should use “possible juvenile, intermediate or adult” because the method is not infallible, and in my concept, you do not demonstrate that the clavicles certainly belong to the same taxon .
Line 110. Replace possesses by posses.
Line 163. What do you mean with “ no differences were found in axial or appendicular skeleton characteristics”, you just have shown images of two isolated clavicles and skulls in different degree of preservation.
Lines 174-175. Only the parietal angle distinguishes Metoposaurus from other Triassic temnospondyl taxa?
Lines 191-197. You mention that exceptions can be. So, which are these exceptions? Could you please include a figure of them?
Lines 198-200. Sentence consider revision, as it is currently, it is not clear what you mean.
Lines 204-208. I would totally agree with this consideration, the difference in the growth pattern can be ecological, so, you can have juveniles or subadults, and adults of a represented species growing faster during early stages of development. At least should be considered as a possibility, but you have to prove that the analyzed isolated bones pertain to the same species and if they belong to individuals of the same population or they belong to timely-separated ones, when conditions are proved to have been different. If you could not prove that, you should leave an interrogation to solve in the future along with new discovering. By the way, metoposaurids are characterized by the wide variability in the ornamentation pattern and long bone morphology, but particularly, ornamentation cannot be used as a diagnostic character because it could even be normal genetic variability (acording to Sulej, 2007, which I did not see in the reference list).
Lines 209-210. Why? Which pattern you expected to find in an intermediate form?
Lines 213-216. If that you state here is true, it would be expected that skulls that show for instance the more terrestrial pattern display other features that were allied to terrestrialization in early tetrapods like the reduction of the otic notch and the lateral line, as well as the reduction of size. Did you note that features?
Lines 222-225. Which is the geological (paleoecological) evidence of these different environmental conditions?
Lines 228-230. Which are these conditions? Please, include a brief description here.
Line 240. Which is the geological evidence for rainy and dry seasons in the Late Triassic Krasiejów formation?
Lines 312-316. Indeed, it is possible to have specimens that represent Metoposaurus juveniles and adults from the same population or from different populations if isolated bones (by the way, they are just clavicles?) are reworked from lower levels and were mixed together secondarily (time averaging should be considered). But doubtfully skulls were reworked, most possible is that they are in situ. Therefore, you have to make your inferences based mainly in the skulls, which surely belong to the same population, although there is no information provided about their spatial distribution when they were found. According to Table 4, you studied near 20 skulls and you could have represented three size-constrained categories (obviously, if you can demonstrate that all belong to the same taxon, which is not clear to me). Therefore, it will be good that you can provide a new table or modify table 4 with the addition of a column that specifies which type of ornamentation have each one of the studied skulls and the area of the skull from which it was observed.

Thus, I hope you found the new revisions and comments useful for get an improved, publisheable version of your manuscript.

Best regards,
Graciela Piñeiro

·

Basic reporting

This manuscript is much improved over the previous version. Particularly important is that the authors have focused on one of the possible causes of the bimodality in dermal bone ornamentation among the Krasiejow metoposaurs. Nevertheless, their argument that the cause is ecological still remains a bit weak, and could be bolstered with some discussion of the taphonomy of the actual metoposaur fossils. I attach an article we published on metoposaur bonebed taphonomy for some ideas in this regard.

Thus, I think the manuscript needs minor to moderate revision to be published.

Spencer G. Lucas

My edits of grammar are on the attached pdf, and comments/suggestions are keyed to numbers on the pdf:

1. You should say that among all these possibilities, ecological variation is most plausible.
2. This specimen should be illustrated and discussed a bit more. If it is not Metoposaurus, then it should be identified.
3. What does “ground” mean? Please clarify.
4. We need to know something more about the taphonomy of the bonebed. Are the two dermal bone morphs obviously from two taphonomically different populations? What evidence of transport is there—alignment, abrasion, etc? See the Lucas et al. 2010 PPP analysis of a Triassic metoposaur bonebed for some ideas about metoposaur taphonomy. A discussion of this would either provide support for the ecological separation idea or not. Indeed, the idea that metopo skulls came from Variscan highlands seems difficult to accept—those skulls are too fragile to travel many kilometers in a river and remain intact.
5. What evidence is there to support this?
6. I don’t follow this? How does biting prey = semi-aquatic lifestyle? Don’t aquatic predators bite their prey?
7. What is “Gilgai relief?”

Experimental design

See above

Validity of the findings

See above

Additional comments

See above

·

Basic reporting

In comparison to the earlier submitted version of this manuscript by Antczak & Biodzioch, the presentation of measurement data has improved; separate supplemental files - e.g. xls sheet files or tab-stop-separated data in txt files - would be better, though.

I still think that at least a photographic depiction of all included specimens whose ornamentations have been measured would greatly support the manuscript as a whole (e.g as figure or supplemental figure).
* * *
Experimental design

The authors have chosen to consider the morphotypes as given - based on qualitative morphological criteria/certain diagnostic features - and use additional (morphometric) measures based on ornamentation patterns in order to test whether the observed differences can be defined quantitatively.

There is the risk of circularity (circular reasoning) if the quantitative criteria used for statistic tests merely reflect the qualitative criteria for the prior distinction of groups. The authors should consider this point by using only one or a few (qualitative) morphological criteria for the definition of morphotypes before the quantitative analysis which then should only involve (quantitative) criteria not already used in the definition of the two types.

The authors' way of reasoning and the newly included significance tests (whether specimens of type I and II come from the same or distinct statistic population(s), Table 5) should be more adequately described in the Methods section.

Concerning the authors' question about MANOVA: I think the only limiting factor might be the number of data points per group in relation to the number of groups that shall be distinguished - so it should be possible in this case.
* * *
Validity of the findings

The presence of two morphotypes appears to be sufficiently supported by data. In accordance with earlier reviewer comments, the authors are now more modest/reluctant about the correspondence of clavicle morphotypes and "dimorphism" in other bones.

There is still the problem that the meaning of the two morphotypes (whether ecology is a decisive factor) is not yet well defined because there is
(a) no outgroup or other reference group (e.g. sufficient specimen sample of another temnospondyl species) on which the same kind of morphometric measurements of ornamentation patterns have been carried out,
(b) no clear assocation of clavicles to other bones of the same individual,
(c) no other additional (e.g. palaeoenvironmental, geochemical, histological) data.

Because such an independent line of evidence is lacking, I'm still not yet convinced by the interpretation of these morphotypes as ecomorphotypes - despite the longish argumentation in the Discussion chapter where the authors try to rule out other ways of interpretation.

I think (in accordance with PeerJ publishing policies) it would inappropriate to demand such additional/indepedent supporting evidence but Abstract, Discussion and Summary should arguably reflect the remaining uncertainty somewhat better.
* * *
Additional comments

The naming of the subheadings has improved, is now reflecting the reasoning/stucture of this approach somewhat better.

Plots: I still think that the convex hulls should include all data points of the respective subgroup/morphotype - no matter the overlap between groups or completeness of the specimens (would be more honest). Otherwise better leave convex hulls out entirely. See my comment in the annotated pdf of the first review.

The ternary plot in Fig. 7 is a nice solution for such kind of compositional data.

The points of concern named above are in agreement with a moderate revision.

Best regards, Michael Buchwitz, 15th February 2018

---

## Round 0.3 · Minor Revisions

Dear authors,

The manuscript has been improved but two important facts remain, in my concept, doubtful:

1-The taxonomic assignation of the clavicles to M. krasiejowensis- The reviewers and I have insisted in that the accuracy of the taxonomic assignation must be guaranteed. But I still do not see a convincing character that allies the isolated clavicles to the skulls. I can understand that all the studied clavicles can be from just one species, as they fall together in the analysis, but this fact is not accurate to say that they belong to M. krasiejowensis. See what important is this, that you based all your work on the clavicle ornamentation to demonstrate the existence of two morphotypes that perfectly could represent two populations. Moreover, a very recent work on M. krasiejowensis humerus histology (Teschner et al., 2018) also found two morphotypes in a fashion very similar to that developed in your manuscript. Authors conclude that there are two populations separated by space and time or there is a sexual dimorphism in this taxon. Intriguingly, this looks perfectly similar to that found by Sulej (2002) when studied Metoposaurus diagnosticus, where according to this author, it is possible to find also two subspecies that differ basically in the length of the prepineal part of the parietal suture. So, I think that the later can be an important character that should be described from the 13 skulls that you used in this study. As I requested previously, it will be necessary to include a schematic drawing of the skulls showing the position of the pineal foramen, but you said that it is covered by sediment in all the 13 specimens. That is very bad because if so, how you know that they belong to one or another taxon? It is true that apparently M. diagnosticus is only found in Germany and M. krasiekpwensis is the Polish form, but if you cannot revise the only two characters that appear to distinguish one from another species, you are only using the geographic precedence without the character-state revision. You must be very explicit about this issue in the article and tell us how you discovered that the 13 examined skulls belong to M. krasiejowensis instead to M. diagnosticus. You also have to include a sentence that explains why you consider that the clavicles that you analyzed belong to M. krasiejowensis (is it just allied to the presence of only one species in Poland?), once you do so, all would be fine to me.

2-Concerning taphonomy, it is very important that you take into account that all reworked materials can be removed from older units or from the same formation. Clavicles can be also reworked more than once and that can explain their bad preservation. Therefore, the clavicles that you studied could have been deposited in older levels than those which provided the skulls, which seem to have a better preservation.

I hope you consider these suggestions to give consistency to your manuscript and then you can submit it for approbation.

Best regards,
Graciela Piñeiro

---

## Round 0.4 · Minor Revisions

Dear authors,

I am very pleased to have seen a so improved manuscript this time, especially the strengthen argumentation to support your thoughts. I am happy to note that.

Therefore, we are close to obtain the final version of your publishable article. You have to revise carefully all my suggestions included in the annotated pdf, attached to this letter. In particular, you should consider my recommendations about the ontogenetic hypothesis, as it may be close related to your preferred one about environmental adaptations and heterochrony. I strongly recommend that you include a new figure, showing the size of the revised clavicles and the pattern to which you assigned each of them.

Finally, you must give attention to the reference list, complete it as I have marked, and take a look on the PeerJ style for consistency.
I wish that you find useful my suggestions, and accept them to improve even more your nice study. I very look forward to see the revised version soon.

Best wishes,
Graciela Piñeiro

---

## Round 0.5 · Minor Revisions

Dear authors,

You have made several advances to reach the final version of your manuscript, but some work remains, mainly grammar issues and some changes for consistency. Please, see the suggested changes at the annotated pdf attached to this letter. Pay attention to the reference list please, I found several papers with problems and a lot not cited in the text.

I look forward to see the next revised version of this interesting manuscript very soon (which I hope will be finally, the acceptable one).

With my best regards,
Graciela Piñeiro

---

## Round 0.6 · Minor Revisions

Dear authors,

We are almost there. Please, see my modifications and corrections in the attached pdf file, particularly those about the intraspecific/ecological variation. Please, be careful to include all the recommended corrections and make a revision of the complete Ms. before re-submit. See that you made some other new typewriting errors when fixing those previously remarked. Revise please the reference list once more; I found at least one article that was not included in the list.

Make all these improvements and resubmit, in such case you do not agree with my suggestions, please explain the reasons in your rebuttal letter.

Best wishes,
Graciela Piñeiro

---

## Round 0.7 · accepted · Accept

Dear authors,

I am very glad to see that you finally arrived to the acceptable version of your interesting manuscript. I really think that this research will be the beginning (the start point) to a larger, possibly interdisciplinary study of Metoposaurus. A deeper taphonomic analysis should be performed to know better the postdepositional events that produced the observed arrangement of the fossils that could belong to the purposed ecological populations. It should be also analyzed the great ontogenetic variability observed in the postcranial bones assigned to Metoposaurus diagnosticus krasiejowensis (cf. Sulej, 2007), and verify if there can be a correlation to the ecotypes that you are describing. Thus, this will be a useful article for encourage a review of the Metoposaurus materials already found and hopefully, to give a new perspective for the new specimens (maybe more completely preserved) that could be discovered in the future.

Congratulations!

With my best regards,
Graciela Piñeiro

#